# CDBridge: A Cross-omics Post-training Bridge Strategy for Context-aware Biological Modeling

**Chang Yu**[1,*] **Siyuan Li**[1,2,*]**, Zicheng Liu**[3]**, Jingbo Zhou**[1,2]**, Xianglong Guo**[3]**,**
**Kai Yu**[1]**, Yuqing Zhou**[1]**, Ken Li**[4]**, Zelin Zang**[1]**, Zhen Lei**[5,6,7,†]**, Stan Z. Li**[1,†]
[1]Westlake University [2]Zhejiang University
[3]Beijing University of Aeronautics and Astronautics
[4]Nanjing University [5]MAIS, Institute of Automation, Chinese Academy of Sciences
[6]School of Artificial Intelligence, University of Chinese Academy of Sciences
[7]CAIR, Hong Kong Institute of Science and Innovation, Chinese Academy of Sciences
{yuchang, Stan.ZQ.Li}@westlake.edu.cn; zhen.lei@ia.ac.cn

## Abstract

Linking genomic DNA to quantitative, context-specific expression remains a central challenge in computational biology. Current foundation models capture either tissue context or sequence features, but not both. Cross-omics systems, in turn, often overlook critical mechanisms such as alternative splicing and isoform reuse. We present CDBridge, a post-training strategy that unifies pretrained DNA and protein models into a context-aware framework without full retraining. CDBridge operates in two stages: (a) *Seq-context learning*, where a splicing-inspired token merge compresses long genomic regions into isoform-aware representations, and (b) *Env-context learning*, where a conditional decoder injects tissue embeddings to model expression under diverse biological contexts. To benchmark this setting, we introduce GTEx-Benchmark, derived from GTEx and Ensembl, which requires models to capture long-range exon dependencies, resolve isoform reuse, and predict tissue-specific expression levels. Across qualitative and quantitative tasks, CDBridge consistently outperforms prior methods that ignore central dogma constraints or context dependence, offering a scalable and biologically faithful solution for DNA-to-expression modeling.

## 1 Introduction

Understanding how genomic DNA sequences give rise to context-specific expression remains a central challenge in computational biology. Here, context involves two complementary aspects: (1) Sequence context, where non-coding regions regulate expression and splicing, allowing a single gene to produce multiple isoforms; and (2) Environmental context, where tissue type or external conditions drastically alter expression levels, even for identical DNA sequences. Accurate modeling of these processes has broad applications, including disease mechanism discovery (Kahles et al., 2018; Nikom & Zheng, 2023; Ueda et al., 2024), drug safety profiling (Ryaboshapkina & Hammar, 2019), and DNA design for synthetic biology (Chen et al., 2025; Yang et al., 2025).

Despite rapid progress, as shown in Table 1, existing methods fall short of bridging DNA to expression in a truly context-aware manner. For example, single-cell foundation models, such as scGPT (Cui et al., 2024), scFoundation (Hao et al., 2024), capture tissue-specific representations but operate on gene IDs, ignoring the underlying DNA sequence that drives expression. In contrast, specialist sequence-to-expression models like Enformer (Avsec et al., 2021), AlphaGenome (Avsec et al., 2025), and Isoformer (Garau-Luis et al., 2024), attempt to incorporate DNA but typically operate on pre-cropped fragments or average across dynamic isoform usage, failing to capture the system's full complexity. Furthermore, while large-scale sequence foundation models (Zhou et al., 2024; Lin

---

[*]Equal Contribution.
[†]Corresponding author.

Table 1: **Comparison of model capabilities across input modalities and design aspects**. CDBridge is the only framework that supports full cross-omics alignment, tissue-aware reasoning, and expression modeling. General indicates that the model supports various tasks, while Specialist indicates the model is designed for seq2express regression tasks. ✓indicates support, ✗indicates not supported.

| Model | Venue | Type | DNA | RNA | Protein | Central Dogma | Express | Tissue-Aware |
|---|---|---|---|---|---|---|---|---|
| scGPT (Cui et al., 2024) | Nat. Methods | General | ✗ | ✗ | ✗ | ✗ | ✓ | ✓ |
| scFoundation (Hao et al., 2024) | Nat. Methods | General | ✗ | ✗ | ✗ | ✗ | ✓ | ✓ |
| GeneCompass (Yang et al., 2024) | Cell Research | General | ✗ | ✗ | ✗ | ✗ | ✓ | ✓ |
| DNABERT2 (Zhou et al., 2024) | ICLR | General | ✓ | ✗ | ✗ | ✗ | ✗ | ✗ |
| NTv2 (Dalla-Torre et al., 2023) | Nat. Mach. Intell. | General | ✓ | ✗ | ✗ | ✗ | ✗ | ✗ |
| HyenaDNA (Nguyen et al., 2024b) | NeurIPS | General | ✓ | ✗ | ✗ | ✗ | ✗ | ✗ |
| Evo (Nguyen et al., 2024a) | bioXiv | General | ✓ | ✗ | ✗ | ✓ | ✗ | ✗ |
| Evo2 (Brixi et al., 2025) | bioXiv | General | ✓ | ✗ | ✗ | ✓ | ✗ | ✗ |
| CD-GPT (Zhu et al., 2024) | bioXiv | General | ✓ | ✓ | ✓ | ✓ | ✗ | ✗ |
| CaLM (Outeiral & Deane, 2024) | Nat. Mach. Intell. | General | ✓ | ✓ | ✓ | ✓ | ✗ | ✗ |
| LucaOne (He et al., 2024) | Nat. Mach. Intell. | General | ✓ | ✓ | ✓ | ✓ | ✗ | ✗ |
| Enformer (Avsec et al., 2021) | Nat. Methods | Specialist | ✓ | ✗ | ✗ | ✗ | ✓ | ✗ |
| AlphaGenome (Avsec et al., 2025) | bioXiv | Specialist | ✓ | ✗ | ✗ | ✗ | ✓ | ✗ |
| Isoformer (Garau-Luis et al., 2024) | NeurIPS | Specialist | ✓ | ✓ | ✓ | ✗ | ✓ | ✗ |
| **CDBridge (Ours)** | **Ours** | General | ✓ | ✓ | ✓ | ✓ | ✓ | ✓ |

et al., 2022; Ji et al., 2021; Zhou et al., 2023; Nguyen et al., 2024b; Dalla-Torre et al., 2023), such as Evo (Nguyen et al., 2024a), LucaOne (He et al., 2024), have advanced DNA and protein modeling across the central dogma, they primarily target qualitative tasks. Consequently, the quantitative nature of expression, which is the ultimate determinant of phenotype, remains largely unaddressed. This gap raises a fundamental question: *How can we map the whole DNA sequence to context-aware quantitative expression?*

Answering this question requires addressing two key challenges: (1) **Sequence length mismatch**, as genes often span hundreds of kilobases while their protein products consist of only a few hundred amino acids; and (2) **Context mapping ambiguity**, since alternative splicing and isoform reuse create inherently one-to-many relationships between DNA and proteins.

To overcome these challenges, we propose CDBridge, a context-aware post-training bridge strategy that unifies pretrained DNA and protein models within a single framework. Unlike prior approaches, CDBridge integrates both sequence-level and tissue-level contexts, enabling simultaneous qualitative and quantitative modeling. The framework proceeds in two stages: (1) Seq-context learning, where a cross-omics connector with cross-attention maps DNA embeddings to protein representations, supported by a splicing-inspired adaptive token merge that selectively compresses non-informative regions while preserving functional signals; and (2) Env-context learning, where a conditional decoder injects tissue embeddings to model tissue-specific expression, selectively activating Stage 1 outputs under given contexts.

To rigorously evaluate central dogma modeling, we introduce GTEx-Benchmark, constructed from GTEx and Ensembl. In contrast to existing benchmarks like Enformer or Isoformer, GTEx-Benchmark forces models to resolve long-range dependencies by identifying critical exons across vast genomic distances, managing exon reuse across multiple isoforms, and predicting tissue-specific expression levels. This creates a challenging and biologically faithful evaluation for central dogma modeling.

Our contributions are threefold:

- *A context-aware, two-stage bridge strategy* that enables cross-omics alignment with minimal paired supervision, supporting both qualitative functional tasks (*e.g.*, protein segmentation) and quantitative expression-level prediction under varied contexts.

- *An adaptive token-merge mechanism* that mimics biological splicing by selectively merging genomic sequences, reducing length disparity and highlighting informative regions for efficient and interpretable modeling.

- *A new benchmark (GTEx-Benchmark)* for tissue-aware central dogma modeling, covering both qualitative cross-omics alignment and quantitative tissue-specific prediction tasks.

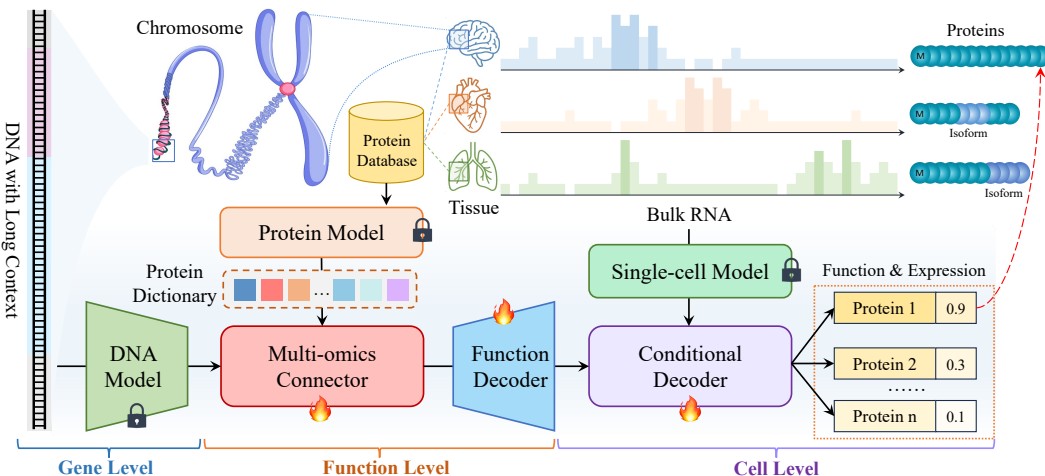

Figure 1: **Overview of the CDBridge framework** for context-aware cross-omics modeling of the central dogma. CDBridge operates in two post-training stages built on frozen DNA and protein foundation models, which consists of two stages: **(1)** *Seq-context learning*, which identifies informative regions from long genomic sequences and maps them to protein-related functional representations. **(2)** *Env-context learning*, a conditional decoder that incorporates tissue embeddings to compose functional features into tissue-specific representations for gene expression prediction.

## 2 METHOD

We introduce CDBridge, a context-aware two-stage post-training framework that bridges single-omics DNA and protein foundation models under the guidance of the central dogma. By implicitly leveraging RNA as a biological mediator, CDBridge aligns cross-modal representations through both sequence-level interaction and tissue-specific context modeling. A left-to-right overview of the architecture is illustrated in Figure 1. Then we introduce our framework from stage 1 to stage 2.

### 2.1 STAGE 1: MULTI-OMICS CONNECTOR FOR LONG-SHORT RANGE BRIDGING

The first stage of CDBridge tackles two core challenges in cross-modal representation alignment: (i) the intrinsic mismatch in sequence length, where a full-length DNA sequence ($\sim 10^4$ tokens) encodes one or more relatively short protein sequences ($\sim 10^2$ tokens); and (ii) the semantic gap between modality-specific pretraining objectives: DNA embeddings capture genome-wide contextual signals, while protein embeddings focus on functional amino acid chains from localized coding regions, which can be seen in the Figure 2(a).

**Framework.** To bridge these gaps, we design a seq-context-aware cross-omics connector that projects the full DNA embedding space into a functionally meaningful protein space using a cross-attention mechanism. Let $\mathbf{X}_{\text{DNA}} \in \mathbb{R}^{L \times d}$ be the DNA embedding sequence produced by a frozen DNA encoder, where $L$ is the token length and $d$ the embedding dimension. We further introduce a learnable token dictionary $\mathcal{T}_{\text{prot}} \in \mathbb{R}^{M \times d}$, initialized from $k$-means clustered protein embeddings across the training dataset. These tokens serve as prototypes, which are treated as keys and values in the cross-attention module:

$$\text{Attn}(\mathbf{X}_{\text{DNA}}, \mathcal{T}_{\text{prot}}, \mathcal{T}_{\text{prot}}) = \text{softmax}\left(\frac{\mathbf{X}_{\text{DNA}}\mathcal{T}_{\text{prot}}^{\top}}{\sqrt{d}}\right)\mathcal{T}_{\text{prot}}.$$

**Token Compression via Merge and Recover.** To efficiently handle the long-range dependencies in genomic sequences, we propose a biologically inspired token compression strategy based on ToMe (Bolya et al., 2023), mimicking transcript splicing mechanisms. This technique allows us to focus computation on functionally relevant regions by adaptively merging non-critical tokens, thereby reducing sequence length while preserving semantic fidelity.

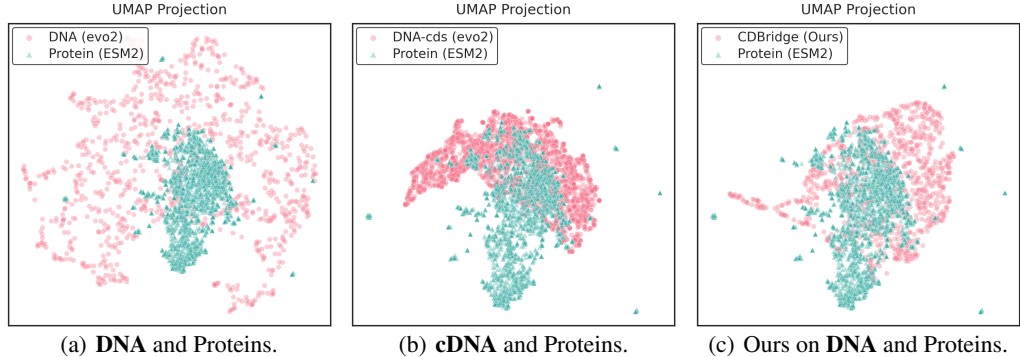

(a) **DNA** and Proteins.    (b) **cDNA** and Proteins.    (c) Ours on **DNA** and Proteins.

Figure 2: **Illustration of the challenges** in aligning DNA and protein representations across modalities. (a) Even with advanced models such as Evo2 and ESM2, alignment remains challenging due to the inherent one-to-many mapping between long DNA sequences and their shorter protein counterparts. (b) Manually segmenting DNA into coding regions reduces input ambiguity, but fails to resolve the representation gap, as existing DNA models are pretrained on full-genome data lacking isoform-specific supervision. (c) By introducing an adaptive token-merging strategy, CDBridge effectively reduces the modality gap and enhances alignment between DNA and protein embeddings.

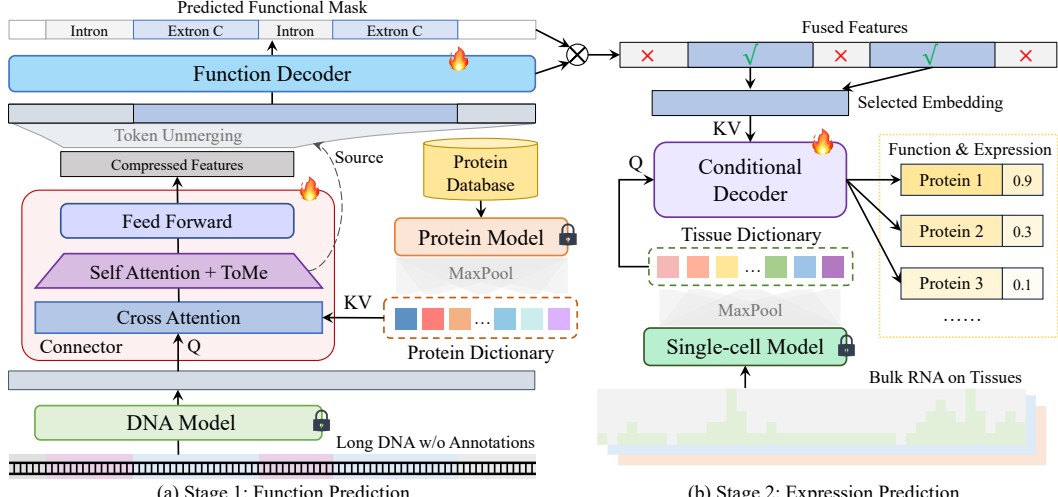

(a) Stage 1: Function Prediction    (b) Stage 2: Expression Prediction

Figure 3: **Two-stage training pipeline of CDBridge**. **(a) Stage 1**: Taking the raw DNA sequences with long contexts as input, whose embedding features are extracted from an existing DNA foundation model (Nguyen et al., 2024a), the multi-omic connector aggregates protein knowledge with cross-attention from the protein dictionary and compresses the fused features with ToMe-Attention (Bolya et al., 2023) while the function decoder predicts the protein functions with token-wise masks. **(b) Stage 2**: After selecting the fused embedding with the predicted masks, the conditional decoder achieves cell-level prediction of expressions of certain proteins with the tissue dictionary.

Given a DNA embedding sequence $\mathbf{X}_{\mathrm{DNA}} = [\mathbf{x}_1, \ldots, \mathbf{x}_L] \in \mathbb{R}^{L \times d}$, we begin by randomly partitioning the token indices into two disjoint sets, $A$ and $B$. For each token $i \in A$, we compute cosine similarity with all tokens $j \in B$ and identify its most similar partner:

$$j^*(i) = \arg\max_{j \in B} \frac{\langle \mathbf{x}_i, \mathbf{x}_j \rangle}{\|\mathbf{x}_i\| \cdot \|\mathbf{x}_j\|}.$$

If the similarity exceeds a threshold $\tau$, the pair $(i, j^*(i))$ is selected for merging. $\tau$ is determined by the pre-set merge ratio, which is randomly sampled from the Gaussian distribution for each input during training. The two tokens are merged via direct averaging:

$$\tilde{\mathbf{x}}_i = \frac{1}{2}(\mathbf{x}_i + \mathbf{x}_{j^*(i)}),$$

where the index $i$ as the surviving token, discarding $j^*(i)$. This yields a compressed sequence $\tilde{\mathbf{X}}_{\text{DNA}} = \{\tilde{\mathbf{x}}_i \mid i \in \mathcal{S}\} \in \mathbb{R}^{L' \times d}$ with $L' < L$. We maintain a mapping function $\pi : \{1, \ldots, L\} \to \mathcal{S} \cup \{\text{NULL}\}$, where $\pi(l) = l$ if token $l$ survives, and $\pi(l) = i$ if token $l$ was merged into token $i$. In the unmerge phase, each discarded token is reconstructed by assigning it the embedding of its surviving partner. The reconstructed sequence $\hat{\mathbf{X}}_{\text{DNA}} = \{\hat{\mathbf{x}}_1, \ldots, \hat{\mathbf{x}}_L\}$ is obtained as:

$$\hat{\mathbf{x}}_l = \begin{cases} \tilde{\mathbf{x}}_l & \text{if } \pi(l) = l \quad \text{(token survived)}, \\ \tilde{\mathbf{x}}_{\pi(l)} & \text{if } \pi(l) \neq l \quad \text{(token was merged)}, \end{cases}$$

which is then processed by a lightweight Transformer decoder to predict functional regions. This *merge-and-recover* procedure can be interpreted as a variant of MAE(He et al., 2022), where the masked regions are adaptively selected based on inter-token similarity. Compared to standard MAE, this strategy is saliency-aware, preserves positional alignment, and enables token-level supervision, making it especially well-suited for genomic sequences where functional signals are sparse and localized. The alignment results are shown in Figure 2. It can be seen that our method shows a better alignment for the global DNA embeddings and the protein embeddings, even exceeds the pre-cropped cDNA and protein embeddings.

## 2.2 STAGE 2: CONDITIONAL DECODER FOR CONTEXT-AWARE EXPRESSION MODELING

While Stage 1 focuses on capturing the structural alignment between DNA and protein through token-level interactions, Stage 2 addresses the variability of gene expression across diverse cellular environments. Specifically, this stage introduces a tissue-aware conditional decoder designed to model the variability of gene expression, capturing the regulatory plasticity that arises from these contextual factors.

**Tissue Dictionary as Conditional Context.** To incorporate relevant biological context, we construct a *Tissue dictionary* $\mathcal{T}_{Envir} \in \mathbb{R}^{C \times M \times d}$ by leveraging a single-cell foundation model(Cui et al., 2024). Here, $C$ represents the number of tissue types, $M$ is the pre-set number of cell tokens, and $d$ denotes the dimensionality of the embedding. Specifically, bulk RNA expression data is first passed through the single-cell foundation model and then pooled to generate global embeddings represented by $M$ tokens. Each tissue is represented by a vector $\mathbf{t}_c \in \mathbb{R}^{M \times d}$ capturing its cell state. During training, tissue labels are supplied as conditional inputs for expression inference, enabling the model to learn differential regulation patterns across distinct biological environments.

**Avoiding Information Leakage from the Context.** The tissue embedding is constructed via mean pooling over approximately 19k genes, without isolating expression values of the target gene or its neighbors. This aggregation effectively dilutes individual gene-level signals. To further verify this, we conducted a control experiment in which the model was trained and tested using only the tissue embedding as input (excluding DNA features). In this setting, the $R^2$ value dropped to nearly zero (see Table 4), confirming that the tissue embedding serves solely as a conditioning signal rather than as an independent predictive feature.

**Conditional Decoder Architecture.** The conditional decoder is a Transformer module that takes the tissue vectors as queries and performs cross-attention over the compressed DNA representations $\tilde{\mathbf{X}}_{\text{DNA}}$ obtained from Stage 1. The output of the Decoder includes $M$ tokens, each representing the candidate isoform-related protein embeddings, which is formulated as:

$$\{\hat{\mathbf{p}}_m\}_{m=1}^M \sim p\left(\{\mathbf{p}_m\}_{m=1}^M \mid \tilde{\mathbf{X}}_{\text{DNA}}, \mathbf{t}_c\right).$$

The decoder outputs enable two types of predictions: (1) isoform-aware protein embeddings, enabling regularization through contrastive loss, and (2) a scalar regression output estimating the quantitative expression level of the target protein under tissue condition $c$. This dual objective ensures both qualitative semantic alignment and quantitative expression estimation. Unlike typical expression models that treat cells as unordered gene sets, our decoder preserves gene-level context and sequence semantics while modeling tissue-dependent effects, yielding fine-grained and generalizable predictions.

Table 2: Expression prediction performance with $R^2(\uparrow)$ and Spearman($\uparrow$) across five specific tissues, along with averaged results over the full dataset. The best results in **bold**. Models are grouped into: (i) auxiliary sequence-only baselines, (ii) specialist expression baselines, and (iii) our cross-omics bridge. Isoformer (Official) relies on a TSS-aligned data setting and is thus not directly comparable to our unaligned, long-sequence protocol.

| Model | Brain | | Heart | | Kidney | | Liver | | Stomach | | Average | |
|---|---|---|---|---|---|---|---|---|---|---|---|---|
| | $R^2$ | Spear | $R^2$ | Spear | $R^2$ | Spear | $R^2$ | Spear | $R^2$ | Spear | $R^2$ | Spear |
| *Auxiliary Sequence-only Baselines* | | | | | | | | | | | | |
| DNABERT2 (Zhou et al., 2024) | -0.004 | 0.317 | -0.005 | 0.304 | -0.004 | 0.317 | -0.001 | 0.328 | 0.001 | 0.333 | -0.004 | 0.317 |
| NTv2 (Dalla-Torre et al., 2023) | -0.012 | 0.238 | -0.003 | 0.317 | -0.135 | 0.176 | -0.023 | 0.291 | 0.005 | 0.306 | -0.012 | 0.289 |
| Evo2-7B (Brixi et al., 2025) | 0.021 | 0.324 | 0.018 | 0.318 | 0.024 | 0.328 | 0.017 | 0.312 | 0.023 | 0.325 | 0.021 | 0.324 |
| LucaOne (He et al., 2024) | 0.006 | 0.320 | -0.001 | 0.300 | 0.007 | 0.324 | -0.003 | 0.314 | 0.002 | 0.318 | 0.001 | 0.309 |
| *Specialist Expression Baselines* | | | | | | | | | | | | |
| Enformer (Avsec et al., 2021) | 0.139 | 0.124 | 0.133 | 0.118 | 0.117 | 0.092 | 0.127 | 0.122 | 0.124 | 0.108 | 0.127 | 0.122 |
| AlphaGenome (Avsec et al., 2025) | 0.234 | 0.442 | 0.260 | 0.404 | 0.229 | 0.380 | 0.221 | 0.438 | 0.242 | 0.410 | 0.248 | 0.438 |
| Isoformer (Official) (Garau-Luis et al., 2024) | 0.505 | – | 0.525 | – | 0.560 | – | 0.530 | – | 0.515 | – | 0.530 | 0.720 |
| Isoformer (w/o TSS Align.) (Garau-Luis et al., 2024) | -0.328 | 0.301 | -0.303 | 0.269 | -0.366 | 0.264 | -0.268 | 0.312 | -0.291 | 0.278 | -0.315 | 0.309 |
| *Proposed Cross-omics Bridge Model* | | | | | | | | | | | | |
| **CDBridge (Ours)** | **0.421** | **0.708** | **0.346** | **0.657** | **0.327** | **0.594** | **0.382** | **0.631** | **0.410** | **0.673** | **0.387** | **0.618** |

## 3 EXPERIMENTS

We evaluate CDBridge across multiple biologically grounded tasks to assess its performance, generalizability, and interpretability.

### 3.1 DATASET CONSTRUCTION.

We construct the GTEx-Benchmark based on the GTEx v8 resource (Consortium, 2020), which provides matched genomic, transcriptomic, and proteomic annotations across 40 human tissues. For each protein-coding gene, we retrieve the DNA sequence and corresponding protein sequence from Ensembl (Cunningham et al., 2022), and pair them with tissue-specific RNA expression values and protein function annotations. We utilize a strict split of 80% training, 10% validation, and 10% testing based on gene IDs to prevent data leakage. To ensure sequence manageability, genes with DNA sequences longer than 200k base pairs are excluded. These ultra-long genes constitute only a small long-tail portion (around 2% of genes in our statistics). The resulting dataset enables evaluation on a wide range of biologically meaningful tasks, including Tissue-conditioned protein expression prediction, Coding region segmentation, and Isoform-level protein retrieval. More details are illustrated in the Appendix A.1.

**Comparison with other methods.** Table 1 presents a comparison of representative models across input modalities, biological reasoning capabilities, and task types. *DNABERT-2* (Zhou et al., 2024) is a single-omics model trained exclusively on DNA sequences, focusing on sequence-level representations. *Evo2* (Brixi et al., 2025) incorporates aspects of the central dogma during training, enhancing sequence modeling, yet it remains fundamentally single-omics. Building upon these, *LucaOne* (He et al., 2024)supports multi-omics inputs spanning DNA, RNA, and protein, offering broader coverage. However, these foundational sequence models lack the capacity for tissue-conditioned reasoning or quantitative expression prediction. In contrast, specialist models such as *Enformer* (Avsec et al., 2021) and *Isoformer* (Garau-Luis et al., 2024) are explicitly designed for expression prediction. However, they rely on fixed-dimension output heads and therefore structurally cannot perform zero-shot prediction on unseen tissues without retraining new heads. Concurrently,

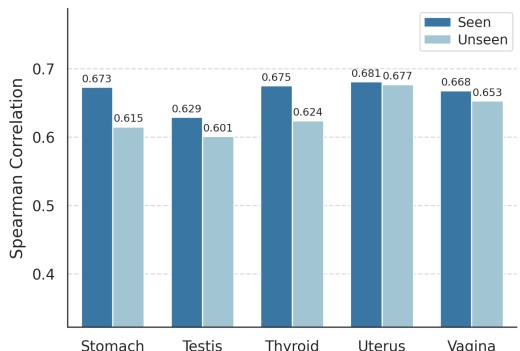

Figure 4: Spearman correlation of gene-expression prediction across five tissues. Each bar reports the Spearman correlation between predicted and ground-truth isoform-level expression in a specific tissue. Model performance on unseen tissues closely mirrors that on tissues observed during training, indicating robust cross-tissue generalization and suggesting that the learned tissue embeddings capture transferable regulatory patterns rather than overfitting to specific training tissues.

AlphaGenome (Avsec et al., 2025) is designed for nucleotide-level quantitative prediction from long DNA contexts across multiple omics. Nevertheless, it emphasizes static averaged outputs for each DNA input, overlooking tissue-dependent variations. Uniquely, *CDBridge* enables comprehensive cross-modal alignment while performing quantitative expression modeling conditioned on tissue contexts. This capability positions *CDBridge* as a versatile biological foundation model, capable of fine-grained, environment-aware inference across multiple molecular modalities.

## 3.2 TISSUE-AWARE GENE EXPRESSION PREDICTION

We evaluate CDBridge on the challenging task of tissue-conditioned isoform-level expression prediction using the GTEx dataset, comparing it against general representative sequence-only models (auxiliary sequence baselines) and specialist models designed for expression prediction. The features of the auxiliary sequence baselines are frozen and followed by a trainable layer for expression regression. Two evaluation settings are considered: (i) seen-tissue (Table 2), where tissues appear during training but test DNA sequences are unseen; and (ii) unseen-tissue (Figure 4), where entire tissue types as well as the test DNA sequences are held out to simulate zero-shot generalization. Specifically, we employ a leave-tissue-out protocol: the model is trained on 90% of tissues and evaluated exclusively on genes from the 10% held-out, unseen tissue types.

From Table 2, sequence-only models such as DNABERT-2 and Evo2 show limited performance since they lack explicit modeling of tissue-specific regulatory signals. While Enformer, AlphaGenome, and Isoformer incorporate expression prediction, they rely on tissue-specific classifiers trained solely on DNA inputs, limiting their ability to generalize to unseen tissues. In contrast, CDBridge consistently achieves superior $R^2$ and Spearman scores across diverse tissues, with particularly strong gains in contexts where regulation is highly tissue-dependent. Figure 4 further demonstrates that CDBridge sustains robust performance even under unseen-tissue conditions. It is a setting unsupported by Enformer or Isoformer, validating its ability to capture transferable context patterns through its two-stage, context-aware architecture.

## 3.3 CROSS-OMICS DOWNSTREAM TASK.

We evaluate the multi-omics representation capacity of CDBridge through three challenging downstream tasks: coding region segmentation, isoform retrieval, and DNA–protein association. Full protocols are provided in Section A.2, and results are summarized in Table 3.

*Coding Region Segmentation.* This task evaluates whether the model can identify which DNA segments code proteins, a key step for genome annotation and clinical variant interpretation. As shown in Table 3, single-omics models such as DNABERT-2 and Evo2 perform relatively poorly due to their inability to disambiguate coding signals from long genomic sequences, while LucaOne benefits from multi-omics embeddings to achieve better alignment. CDBridge outperforms them, demonstrating its ability to capture Isoformer-related protein signals for fine-grained token-level tasks.

*Isoform Retrieval.* Given a set of candidate isoforms, this task assesses the ability to retrieve the top-$K$ activated isoforms from DNA sequences with different conditions. It is important for disease mechanism discovery. As expected, unimodal DNA models perform poorly. LucaOne, while benefiting from multi-omics embeddings, still provides only moderate performance due to the absence of fine-grained isoform-level modeling. In contrast, CDBridge achieves the best results, leveraging cross-modal alignment and tissue-aware conditioning to capture tissue-specific isoform usage patterns accurately.

*Central Dogma.* This binary classification task evaluates if a DNA-protein pair is functionally associated, relevant to disease mechanism discovery and drug target identification. The Central Dogma dataset from LucaOne is used. DNABERT-2 lacks the capacity to incorporate protein information, yielding weak performance. Evo2 and LucaOne performs better, benefiting from joint modeling, but lack task specialization. CDBridge achieves the highest performance by leveraging multi-omics conditioning and structured alignment.

Table 3: Comprehensive evaluation across three biological tasks (best **bold**, second-best underlined).

| Model | Coding Region Segmentation | | | Isoform Retrieval | | Central Dogma | | |
|---|---|---|---|---|---|---|---|---|
| | Acc ↑ | AUC ↑ | F1 ↑ | Acc ↑ | MRR ↑ | Acc ↑ | AUC ↑ | F1 ↑ |
| Random | 0.617 | 0.269 | 0.002 | 0.010 | 0.181 | 0.503 | 0.499 | 0.502 |
| *Single-omics General models* | | | | | | | | |
| NTv2-500M (Dalla-Torre et al., 2023) | 0.814 | 0.529 | 0.134 | 0.145 | 0.229 | 0.572 | 0.597 | 0.415 |
| DNABERT-2 (Zhou et al., 2024) | 0.851 | 0.612 | 0.382 | 0.132 | 0.227 | 0.574 | 0.598 | 0.482 |
| Evo2 (Brixi et al., 2025) | 0.993 | 0.848 | 0.597 | 0.191 | 0.278 | 0.672 | 0.725 | 0.518 |
| *Muti-omics General models* | | | | | | | | |
| LucaOne (He et al., 2024) | 0.993 | 0.859 | 0.613 | 0.259 | 0.354 | 0.714 | 0.767 | 0.545 |
| **CDBridge (Ours)** | **0.995** | **0.993** | **0.635** | **0.337** | **0.436** | **0.742** | **0.792** | **0.568** |

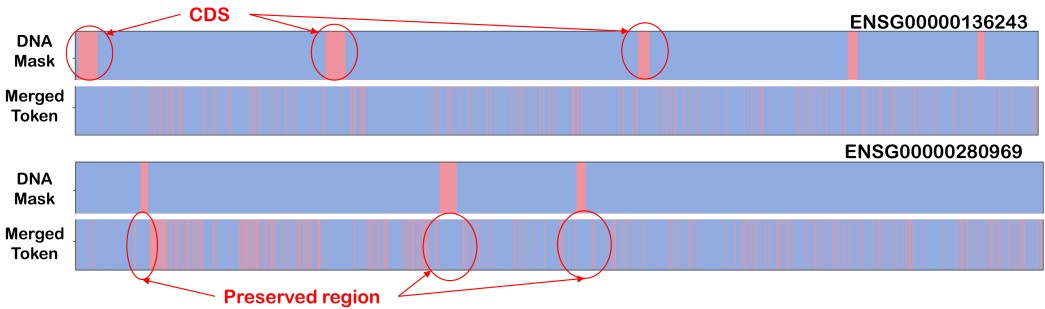

Figure 5: **Token merging aligns with functional regions.** The merging process selectively retains tokens associated with coding regions, while non-coding segments are predominantly merged. This pattern arises without explicit exon masks in the loss, indicating that CDBridge learns to allocate computation to biologically salient regions in a data-driven and interpretable manner.

## 3.4 INTERPRETABLE ANALYSIS

*Adaptive merge salient regions.* We visualize the token merging behavior by overlaying the merging heatmap with known exon/intron annotations. The experiment is conducted on held-out test samples, where we visualize the top 100 most frequently reused tokens during the adaptive token merging stage. As shown in Figure 5, without explicit masking for merging, the model consistently retains tokens corresponding to coding regions (exons), while aggressively merging tokens in non-coding regions (introns or intergenic). This behavior suggests that CDBridge effectively allocates computation to biologically significant regions in a data-driven and interpretable manner.

*Tissue-aware activations.* The activated tokens under different conditions are visualized in Figure 6. The first gene sample activates tokens predominantly associated with the first two isoforms, and the activation patterns shift based on tissue type, reflecting expression differences. Similarly, the second gene shows consistent activation in tokens 4–16 across tissues, but with varying intensities. These patterns indicate that CDBridge is capable of modeling the tissue-specific regulatory contexts.

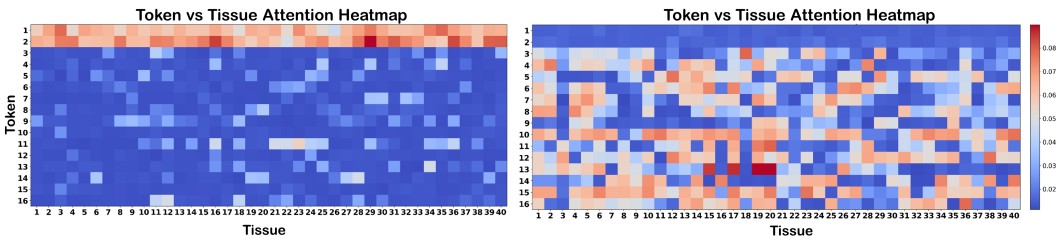

Figure 6: **Tissue-specific token activations.** Each panel displays the token activation magnitude for specific isoforms (rows) across various tissue contexts (columns). CDBridge selectively activates isoform-associated tokens, with varying intensity across tissues, reflecting its capacity to model tissue-aware gene expression across different DNA regions.

Table 4: **Ablation study of CDBridge components.** Each row represents a different configuration of Stage 1 (segmentation) and Stage 2 (expression prediction). ✓ indicates the corresponding component is enabled. Segmentation performance is measured by AUC and F1, while expression prediction is evaluated using $R^2$ and Spearman correlation. Numbers in $(+\Delta)$ show relative improvements over the baseline.

| Stage 1 | | | Stage 2 | Segmentation | | Expression | |
|---|---|---|---|---|---|---|---|
| ToMe Attn. | Fixed Clust | Learned Clust | Tissue Clust | AUC ↑ | F1 ↑ | $R^2$ ↑ | Spear ↑ |
| ✗ | ✗ | ✗ | ✗ | 0.848 | 0.600 | 0.021 | 0.324 |
| ✓ | ✗ | ✗ | ✗ | 0.882(+0.034) | 0.601(+0.001) | 0.205(+0.184) | 0.457(+0.133) |
| ✓ | ✓ | ✗ | ✗ | 0.990(+0.142) | 0.602(+0.002) | 0.212(+0.191) | 0.483(+0.159) |
| ✓ | ✗ | ✓ | ✗ | **0.993**(+0.145) | **0.635**(+0.035) | 0.215(+0.194) | 0.483(+0.159) |
| ✗ | ✗ | ✗ | ✓ | – | – | 0.020(-0.001) | 0.128(-0.196) |
| ✓ | ✗ | ✓ | ✓ | **0.993**(+0.145) | **0.635**(+0.035) | **0.387**(+0.366) | **0.618**(+0.294) |

## 3.5 ABLATION STUDY

To understand the contribution of each component in our two-stage CDBridge framework, we conduct systematic ablation experiments along both segmentation and expression pathways (Table 4). The first stage includes a ToMe module and a protein clustering module, with or without learning. The second stage evaluates the presence of tissue-specific clustering during expression prediction.

In the segmentation task, removing the entire Stage 1 (*i.e.*, no ToMe or protein clustering) leads to a significant drop in performance (AUC = 0.848, F1 = 0.600), matching the Evo2 baseline. Introducing ToMe alone slightly improves segmentation, suggesting the benefit of adaptive compression. Adding a non-learnable protein cluster yields moderate gains, while learning the protein cluster jointly with ToMe provides the best segmentation performance. For the expression task, disabling tissue conditioning results in poor generalization ($R^2$ = 0.020, Spearman = 0.128), confirming that incorporating single-cell models for tissue-aware modeling will not introduce information leakage. Enabling all components achieves a strong improvement ($R^2$ = 0.387, Spearman = 0.618), validating the design of the two-stage architecture.

## 4 RELATED WORK

**Pretrained Biological Language Models.** Recent years have witnessed rapid progress in large-scale pretrained models for biological sequences. On the molecular level, protein language models such as ESM (Rives et al., 2021), Evo (Meier et al., 2021), and AlphaFold (Jumper et al., 2021) have shown impressive performance in protein structure prediction, function classification, and representation learning. On the genomic side, models like Enformer (Avsec et al., 2021) and Nucleotide Transformer (Dalla-Torre et al., 2023) leverage long-range attention to model cis-regulatory elements and gene expression from raw DNA sequences. While these models excel within their own modalities, they are typically trained independently and lack mechanisms for aligning representations across the molecular hierarchy from DNA to protein, limiting their utility in tasks that require reasoning over the central dogma.

**Multi-omics Modeling of the Central Dogma.** To bridge across DNA, RNA, and protein, multi-omics models such as CD-GPT (Zhu et al., 2024), GENA-LM (Ji et al., 2023), Life-Code (Liu et al., 2025), and LucaOne (He et al., 2024) have been proposed to unify biological sequences in a shared embedding space. These models usually decompose input sequences into functional segments (*e.g.*, coding vs non-coding) and attempt modality alignment via paired training. Some focus on *qualitative modeling* like function transfer, while others explore *quantitative prediction* of gene expression. However, most overlook crucial biological context: (1) splicing and regulatory mechanisms often cause one gene to yield multiple proteins, and (2) the same DNA may lead to different expression outcomes depending on the cellular or environmental context. These factors make DNA-to-protein alignment inherently ambiguous and context-dependent, a challenge under-addressed in existing models. In contrast to unified multi-omics foundation models like GENA-LM and LucaOne, which require end-to-end multi-omics pretraining, CDBridge operates as a post-training bridge that augments frozen single-omics DNA and protein encoders with tissue-aware cross-omics reasoning, avoiding expensive retraining while retaining the flexibility of modular single-omics backbones.

**Multimodal Connectors and Bridge Strategies.** Given the high cost of collecting large-scale multimodal datasets, a promising direction is to bridge pretrained single-omics models through lightweight post-training alignment. Existing efforts in general multimodal AI (Radford et al., 2021; Chen et al., 2024; Xue et al., 2024; Li et al., 2023; Alayrac et al., 2022) have shown success in domains like vision-language, but biology poses unique challenges such as extreme sequence length disparities and complex one-to-many mappings (*e.g.*, splicing, RNA editing). Few biological models have explored connector-based strategies that are both context-aware and biologically grounded. In this work, we propose CDBridge, a post-training bridge framework that incorporates both sequence and cellular context, aligning existing DNA and protein models to support realistic, condition-aware biological tasks.

## 5 CONCLUSION

We present CDBridge, a biologically grounded and context-aware framework that bridges single-omics foundation models through a two-stage design inspired by the central dogma. By leveraging ToMe-based token compression to capture isoform-aware coding structures and introducing a conditional decoder to model tissue-specific regulation, CDBridge enables fine-grained, condition-dependent protein expression modeling directly from DNA. Our approach integrates both structural semantics and environmental context, outperforming existing models that either ignore genomic continuity or lack regulatory awareness. CDBridge not only establishes a scalable method for DNA-to-protein reasoning but also opens new avenues for complex biological systems.

**Limitations** Despite its promise, CDBridge still exhibits several constraints. First, the framework relies on high-quality isoform annotations and tissue-resolved expression atlases; incomplete or noisy metadata can propagate errors through both stages. Second, limited by the scarcity of publicly accessible fine-grained contextual data, our conditional decoder currently models tissue context as a categorical variable, leaving unaccounted finer-grained factors such as developmental stage, disease state, and microenvironmental cues. Third, while ToMe compression mitigates sequence length, end-to-end training on whole-genome inputs remains computationally demanding, limiting scalability to large cohorts or non-human genomes with less curated references. Future work could relax the reliance on curated isoform annotations by incorporating transcript assembly or junction-level supervision, and extend the conditional decoder to model continuous or spatially resolved contexts (*e.g.*, developmental time, disease severity, or spatial micro-environments).

### ACKNOWLEDGMENTS

This work was supported by Open Research Fund of The State Key Laboratory of Multimodal Artificial Intelligence Systems (No.MAIS2025064), National Science and Technology Major Project (No.2022ZD0115101), National Natural Science Foundation of China Project (No.624B2115 No.U21A20427), Project (No.WU2022A009) from the Center of Synthetic Biology and Integrated Bioengineering of Westlake University, the Hangzhou Postdoctoral Daily Funding Program (No.103140026582502, 2025) and InnoHK Program.

**Reproducibility Statement.** We have made significant efforts to ensure the reproducibility of our work. Detailed descriptions of the model architecture, training objectives, and evaluation protocols are provided in Sections 3 and 4 of the main text, with additional implementation details, hyperparameters, and dataset statistics included in the Appendix and supplemental materials. Upon acceptance, we will publicly release the full GTEx-Benchmark dataset splits and preprocessing scripts, as well as the CDBridge codebase and pretrained models, to enable end-to-end reproduction of all experiments. This will ensure that other researchers can directly validate our results and extend our framework to new settings.

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

# A GTEx-BENCHMARK

We introduce **GTEx-Benchmark**, a comprehensive dataset designed to evaluate models of the Central Dogma through both qualitative and quantitative tasks. Our goal is to provide a cross-omics benchmark that adheres to biological principles, specifically the DNA → RNA → protein flow—and captures tissue-specific regulatory complexity.

For each protein-coding gene, we extract genomic DNA sequences and their corresponding protein products from Ensembl (Cunningham et al., 2022), and pair them with tissue-resolved RNA expression profiles from GTEx-v8 (Consortium, 2020). Additionally, we include protein-level annotations such as coding region mappings to support functional analysis. This enables two key task types: (1) *qualitative alignment*, such as isoform-level protein retrieval given DNA; and (2) *quantitative prediction*, such as tissue-specific gene expression modeling. To broaden evaluation, we also incorporate the Central Dogma subset from LucaOne (He et al., 2024), which provides curated DNA–protein pairs with explicit alignment.

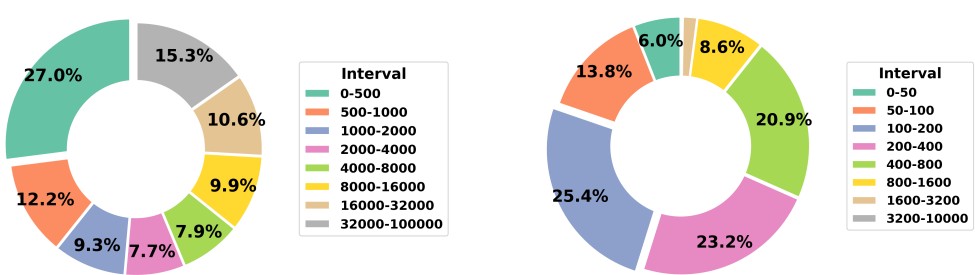

(a) DNA length distribution.    (b) Protein length distribution.

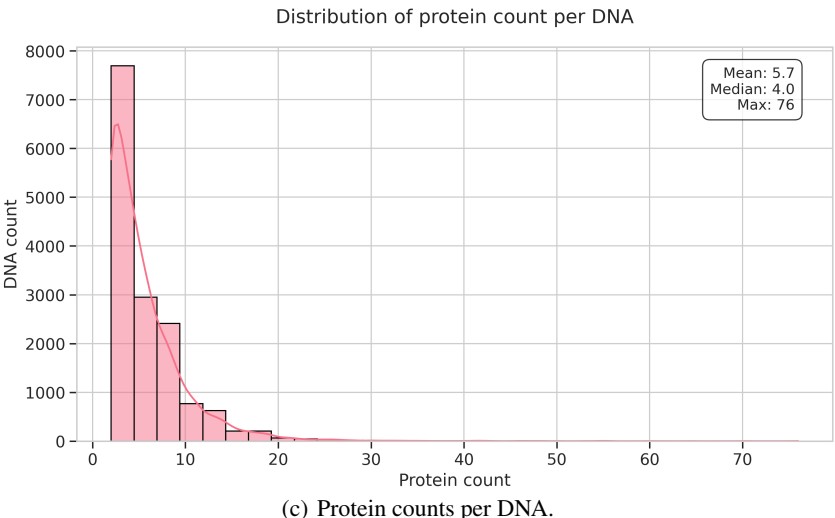

(c) Protein counts per DNA.

Figure 7: Sequence statistics of the GTEx-Benchmark dataset.

## A.1 DATA STATISTICS

*Sequence Distribution.* GTEx-Benchmark comprises over 19,000 human genes, each containing both coding and non-coding DNA regions. Owing to alternative splicing, a single genomic sequence can give rise to multiple transcript variants, resulting in diverse protein isoforms with distinct functional roles. To ensure computational tractability, we exclude genes with DNA sequences longer than 200,000 nucleotides. We divided the dataset into training set, validation set, and test set in the proportions of 80%, 10%, and 10%, respectively. Figure 7 illustrates key statistics of the dataset: (a) the distribution of DNA sequence lengths, (b) the distribution of protein sequence lengths, and (c)

the number of protein isoforms associated with each gene. These figures underscore the substantial sequence and isoform-level heterogeneity that models must handle when reasoning across omics layers.

*Tissue-aware expression preprocessing.* The GTEx project provides gene expression measurements across 49 human tissues, enabling context-dependent modeling. To ensure comparability, we normalize transcript-level expression values using TPM (Transcripts Per Million) and apply a log transformation, $\log(\text{TPM} + 1)$, for numerical stability during training. We define a transcript as *activated* in a given tissue if its TPM exceeds 0.1. Based on this threshold, we compute the number of activated protein isoforms per gene across tissues. As shown in Figure 8, the average number of activated isoforms per gene is approximately five, underscoring the complexity introduced by tissue-specific expression and splicing.

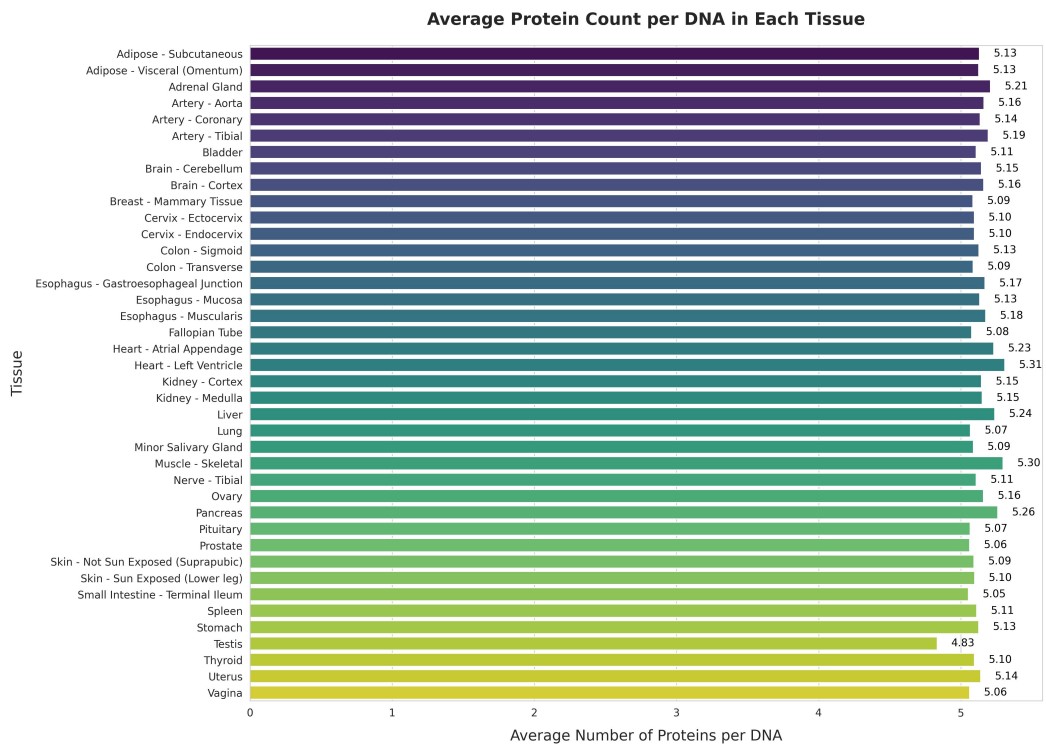

Figure 8: Distribution of the number of activated protein isoforms per gene across tissues in GTEx-Benchmark.

## A.2   EVALUATION PROTOCOL AND DOWNSTREAM TASKS

To evaluate the generalization and cross-modal reasoning capabilities of CDBridge, we consider three biologically grounded downstream tasks: *Coding Region Segmentation*, *Protein Retrieval*, and *DNA–Protein Association Classification*. These tasks are designed to reflect different aspects of the central dogma, ranging from token-level sequence understanding to modality-spanning alignment and functional prediction, and follow the same evaluation setup as in Section 3.

*Coding Region Segmentation.* Each genomic locus is provided in full length (up to 200 kb), and the model predicts whether each nucleotide belongs to a canonical coding sequence. Ground-truth labels are obtained from Ensembl CDS annotations. To reduce label noise at exon–intron junctions, positions within $\pm 3$ bp of splice boundaries are ignored during metric calculation. We evaluate with accuracy, AUC, and F1, where AUC is reported both globally (pooled nucleotides) and averaged across loci, ensuring that performance is not dominated by a few long sequences.

*Isoform Retrieval.* We construct gene–tissue pairs from GTEx by selecting genes with at least two expressed isoforms in a tissue. The tissue is conditioned with scGPT (Cui et al., 2024) and is concatenated with DNA embeddings as input. Each query contains a candidate isoform set, and the

model is asked to rank them according to predicted expression in the given tissue. Ground-truth usage is defined by the median TPM across GTEx donors. Performance is measured by $\mathrm{Acc@3}$, the fraction of cases where the true top isoform falls in the top three predictions, and MRR, the mean reciprocal rank of the top isoform:

$$\mathrm{MRR} = \frac{1}{N} \sum_{i=1}^{N} \frac{1}{\mathrm{rank}_i}, \tag{1}$$

where $\mathrm{rank}_i$ is the predicted rank position of the ground-truth top isoform for the $i$-th gene–tissue pair, and $N$ is the number of evaluation pairs. This protocol reflects practical applications such as pinpointing tissue-dominant isoforms for disease mechanism studies, where ranking a few plausible candidates is more valuable than identifying a single absolute prediction. Compared to standard expression regression, this setup directly tests whether a model can resolve isoform ambiguity under changing cellular environments.

*DNA–Protein Association (Central Dogma Classification).* we adopt the Central Dogma subset from LucaOne (He et al., 2024), which links genomic DNA segments to their translated proteins. Positive pairs are derived from canonical gene–isoform mappings, while negative pairs are generated by sampling proteins from other genes matched by length and chromosome to avoid trivial cues. The dataset contains around 20k pairs with class-balanced sampling. We report accuracy, AUC, and F1. This task requires the model to capture functional correspondence across modalities rather than simple sequence similarity, and is directly relevant to applications such as drug target validation, where identifying functionally matched DNA–protein pairs is critical.

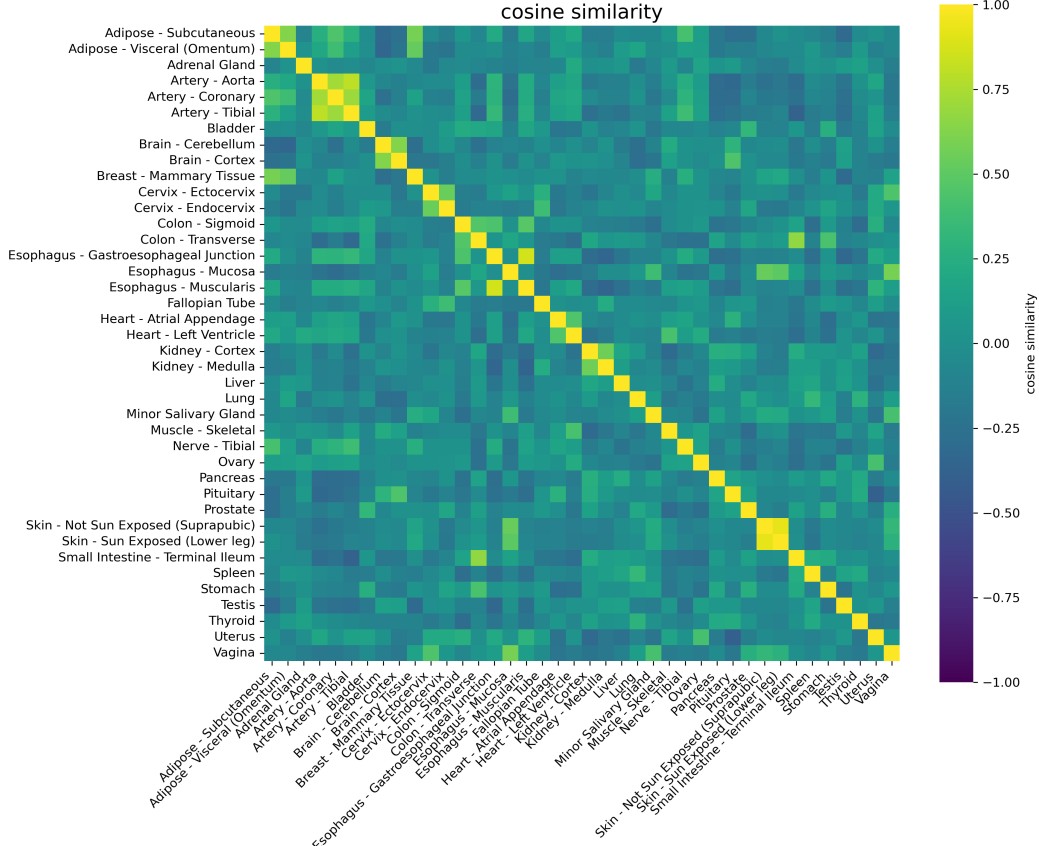

Figure 9: Cosine similarity heatmap of tissue embeddings generated by scGPT (Cui et al., 2024). Related tissues (*e.g.*, arteries) cluster together, suggesting biological consistency. This structure supports the use of scGPT-derived tissue embeddings as conditioning signals in CDBridge, since they preserve biologically meaningful relationships between tissues.

## B DETAILS OF THE METHOD

### B.1 TISSUE EMBEDDING.

To support generalizable cross-tissue prediction, we incorporate tissue embeddings derived from the cell foundation model scGPT (Cui et al., 2024). For each tissue, we extract its representation from scGPT and apply mean-centering across all tissues by subtracting the average embedding across dimensions. This debiasing step enhances inter-tissue contrast and facilitates better context conditioning in downstream tasks. Figure 9 shows the cosine similarity heatmap among the resulting tissue embeddings. As expected, biologically related tissues such as *Artery-Aorta* and *Artery-Coronary*, which belong to the same higher-level anatomical category, exhibit stronger similarity. This structure reveals that scGPT captures semantically meaningful and biologically consistent relationships between tissues, supporting its use as a compact and transferable tissue representation.

### B.2 MERGE RATIO

Merge ratio indicates the percentage of tokens merged during the forward pass. During training, the merge ratio is sampled dynamically for each batch from a clipped Gaussian distribution: $r \sim \mathcal{N}(\mu = 0.375, \sigma = 0.1)$, clipped to the range $[0.25, 0.50]$. We employ this dynamic sampling strategy as a form of structural data augmentation. By forcing the model to perform segmentation and alignment under varying degrees of compression, we prevent it from overfitting to specific sequence strides or fixed resolutions.

### B.3 LOSS FUNCTIONS

CDBridge adopts a two-stage training paradigm. Stage 1 focuses on qualitative token-level alignment between DNA and protein isoforms, while Stage 2 performs quantitative, tissue-aware prediction of protein expression levels. The overall training objective combines classification and contrastive components.

**Stage 1: Multi-omics Connector for Long-Short Range Bridging.** This stage optimizes two complementary losses:

*(1) Multi-label Binary Classification Loss.*To supervise the coding region segmentation, we use a binary cross-entropy loss over the predicted probability of each DNA token coding for any protein isoform. The loss is computed as:

$$\mathcal{L}_{\text{cls}} = \text{BCEWithLogitsLoss}(y_{\text{pred}}, y_{\text{mask}}),$$

where $y_{\text{pred}}$ are the model logits and $y_{\text{mask}}$ is the binary mask indicating coding tokens for each isoform.

*(2) Cross-modal Token-level Contrastive Loss.* To enforce fine-grained alignment between DNA and protein token embeddings, we employ a batch-wise self-supervised contrastive loss inspired by SimCLR(Chen et al., 2020). For each aligned DNA-protein token pair (as defined by the binary mask), we maximize similarity and simultaneously push apart negative DNA samples selected randomly within the same sequence. Specifically:

$$\mathcal{L}_{\text{align}} = -\frac{1}{N}\sum_{i=1}^{N}\log\frac{\exp\left(\text{sim}(z_i^p, z_i^d)/\tau\right)}{\exp\left(\text{sim}(z_i^p, z_i^d)/\tau\right) + \sum_{k=1}^{K}\exp\left(\text{sim}(z_i^p, z_{ik}^{d-})/\tau\right)},$$

where $z_i^p$ and $z_i^d$ are the normalized embeddings of the $i$-th aligned protein and DNA token, $z_{ik}^{d-}$ are randomly sampled negative DNA embeddings, and $\tau$ is a temperature hyperparameter. The similarity function $\text{sim}(\cdot, \cdot)$ is implemented as the dot product of L2-normalized vectors.

The final loss for Stage 1 is the sum of both components:

$$\mathcal{L}_{\text{stage1}} = \mathcal{L}_{\text{cls}} + \lambda_{\text{align}}\mathcal{L}_{\text{align}},$$

where $\lambda_{\text{align}}$ is a hyperparameter controlling the strength of the global alignment.

**Stage 2: Tissue-aware Expression Regression.** In the second stage, the model is trained to predict protein expression levels in a tissue-aware manner. The input is the merged DNA representation from Stage 1, conditioned on tissue embeddings. The regression task supervises quantitative mapping from DNA to protein isoform expression values under different tissue conditions.

*(1) Tissue-specific Regression Loss.* For each protein isoform, we extract its expression value under the corresponding tissue index and regress from the model's output. The loss is formulated as:

$$\mathcal{L}_{\text{expr}} = \frac{1}{B \times K} \sum_{i=1}^{B} \sum_{k=1}^{K} (\hat{y}_{ik} - y_{ik})^2 ,$$

where $B$ is the batch size, $K$ is the number of proteins, $\hat{y}_{ik}$ is the predicted expression, and $y_{ik}$ is the ground-truth expression under the sampled tissue.

*(2) Global Embedding Consistency Loss.* To enhance representational alignment across tissues and protein isoforms, we added a global contrastive loss to enforce the consistency between the pooled DNA embedding and the protein representation. This prevents overfitting to token-level patterns and encourages global biological coherence. We use a symmetric contrastive loss defined as:

$$\mathcal{L}_{\text{global}} = -\frac{1}{B} \sum_{i=1}^{B} \log \frac{\exp\left(\text{sim}(z_i^{\text{dna}}, z_i^{\text{protein}})/\tau\right)}{\sum_{j=1}^{B} \exp\left(\text{sim}(z_i^{\text{dna}}, z_j^{\text{protein}})/\tau\right)},$$

where $z_i^{\text{dna}}$ and $z_i^{\text{protein}}$ denote the pooled (*e.g.*, mean or CLS-token) global embeddings for sample $i$, and $\tau$ is the temperature.

The total loss for Stage 2 training is the sum of both:

$$\mathcal{L}_{\text{stage2}} = \mathcal{L}_{\text{expr}} + \lambda_{\text{global}} \cdot \mathcal{L}_{\text{global}},$$

where $\lambda_{\text{global}}$ is a hyperparameter controlling the strength of the global alignment.

## C  MODEL PARAMETERS AND TRAINING SETUP

CDBridge is built on top of powerful pre-trained single-omics models to facilitate tissue-aware cross-omics bridging. Specifically, we use the 7B Evo2 model (Brixi et al., 2025) as the DNA encoder, the 650M ESM2 model (Lin et al., 2022) for protein representation, and scGPT (Cui et al., 2024) to provide tissue embeddings.

**Training Strategy.** The training process is divided into two stages. In Stage 1, both the encoder and decoder are trainable to learn fine-grained DNA-protein alignment and coding region segmentation. In Stage 2, we freeze the encoder and decoder components from Stage 1 to preserve the learned cross-modal alignment. A new tissue-aware decoder is introduced to regress tissue-specific protein expression levels. This separation ensures that the model retains sequence-level alignment while learning contextual regulation patterns for expression.

**Trainable Components.** CDBridge is built upon pre-trained single-omics models to enable tissue-aware cross-omics prediction. Specifically, we utilize the 7B Evo2 model (Brixi et al., 2025) as the DNA encoder, the 650M ESM2 model (Lin et al., 2022) for protein representation and alignment, and scGPT (Cui et al., 2024) for deriving tissue embeddings. Table 5 summarizes the trainable components and their parameter counts across both training stages. In Stage 2, the encoder and decoder from Stage 1 are frozen to preserve learned alignment, and only the tissue-aware decoder and expression regression layers are updated during training.

## D  MORE ABLATIONS

As shown in Table 6, we present additional ablation results evaluating CDBridge's performance when integrating features from various pretrained models. As shown in the table below, the combination of Evo2 and ESM2 achieves the best overall performance across all three tasks. Additionally,

Table 5: Trainable parameter counts for key modules.

| Module | Params | Notes |
|---|---|---|
| ToMe-based cross-omics encoder | $\sim$40.4M | *Frozen in Stage 2* |
| Transformer decoder (Stage 1) | $\sim$20.7M | *Frozen in Stage 2* |
| Tissue Conditional Decoder (Stage 2) | $\sim$8.3M | Trainable in Stage 2 |
| Classification Head (Stage 1) | $\sim$415K | *Frozen in Stage 2* |
| Regression Layer (Stage 2) | $\sim$513K | Trainable in Stage 2 |

Table 6: More ablation studies and comparison results across GTEx expression prediction and its two sub-tasks, where **bold** denotes the best results.

| Model | Coding Region Segmentation | | | Isoform Retrieval | | GTEx Expression | |
|---|---|---|---|---|---|---|---|
| | Acc ↑ | AUC ↑ | F1 ↑ | Acc ↑ | MRR ↑ | $R^2$ ↑ | Spear ↑ |
| Random | 0.617 | 0.269 | 0.002 | 0.010 | 0.181 | -13.027 | 0.005 |
| NTv2-500M (Dalla-Torre et al., 2023) | 0.814 | 0.529 | 0.134 | 0.145 | 0.229 | -0.012 | 0.289 |
| DNABERT-2 (Zhou et al., 2024) | 0.851 | 0.612 | 0.382 | 0.132 | 0.227 | -0.004 | 0.317 |
| Evo2 (Brixi et al., 2025) | 0.993 | 0.848 | 0.597 | 0.191 | 0.278 | 0.021 | 0.324 |
| LucaOne (He et al., 2024) | 0.993 | 0.859 | 0.613 | 0.259 | 0.354 | 0.001 | 0.309 |
| CDBridge (NTv2, ESM2) | 0.912 | 0.876 | 0.572 | 0.197 | 0.271 | 0.371 | 0.579 |
| CDBridge (DNABERT-2, ESM2) | 0.935 | 0.950 | 0.613 | 0.304 | 0.381 | 0.375 | 0.583 |
| CDBridge (Evo2, ESM3) | 0.993 | 0.989 | 0.616 | **0.339** | 0.434 | 0.382 | 0.607 |
| **CDBridge (Evo2, ESM2)** | **0.995** | **0.993** | **0.635** | 0.337 | **0.436** | **0.387** | **0.618** |

integrating multiple pretrained features clearly improves upon single-modality baselines. Although ESM-3 employs VQ-quantized features incorporating amino acid sequences, protein structures, and functional annotations (thus excelling at the protein retrieval task), it slightly underperforms ESM-2 in CDS segmentation and expression prediction, likely due to reduced compatibility with DNA-sequence-level detail required by these tasks.

Besides, we performed an additional ablation study comparing the performance of using a fixed clustering mechanism versus our learned clustering mechanism within the full Stage 2 model. This comparison validates the necessity of the learned, data-driven approach for optimal performance. The results are summarized in Table 7. The significant performance improvement observed with the learnable mechanism confirms that data-driven cluster assignment is crucial for effectively integrating sequence and tissue context information.

Table 7: Fixed vs. Learned Clustering on GTEx Expression Prediction.

| Clustering Mechanism | $R^2$ | Spearman |
|---|---|---|
| Fixed | 0.305 | 0.472 |
| Learnable | **0.387** | **0.618** |

# E COMPARISON WITH ADDITIONAL SOTA BASELINES

To further validate the effectiveness of CDBridge against state-of-the-art sequence-based models, we conducted a comparison with Borzoi (Linder et al., 2025), a recently proposed successor to Enformer. Similar to Enformer, Borzoi shares the fundamental fixed-output architecture designed for predicting epigenomic tracks across the genome.

As summarized in Table 8, while Borzoi demonstrates improved performance compared to Enformer (refer to main text), it remains significantly below CDBridge on our isoform-level, tissue-conditioned benchmark ($R^2$ of 0.201 vs. 0.387). This performance gap reinforces our observation that fixed-output architectures struggle with the dynamic, context-conditioned modeling required for resolving isoform-specific expression across diverse tissues.

Table 8: Performance comparison with Borzoi on GTEx Expression Prediction.

| Model | $R^2$ | Spearman |
|---|---|---|
| Borzoi (Linder et al., 2025) | 0.201 | 0.336 |
| **CDBridge (Ours)** | **0.387** | **0.618** |

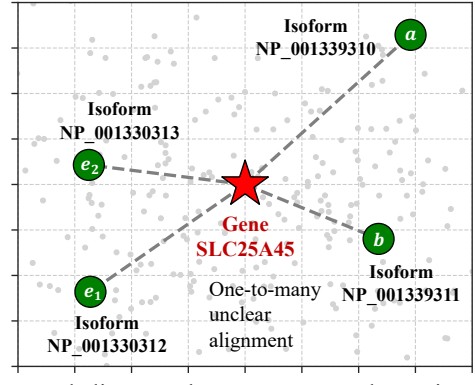
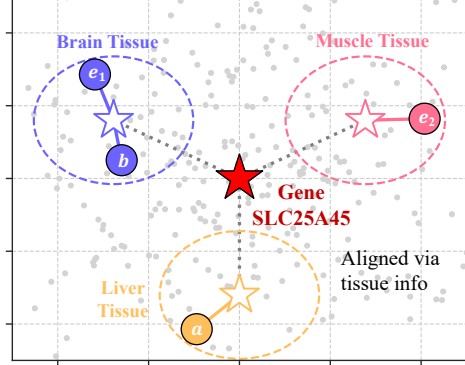

Hard alignment between Gene and Proteins          Tissue-conditioned Alignment

Figure 10: **Tissue context enables meaningful gene–protein alignment.** (Left) Without tissue information, SLC25A45 (red) and its four isoforms (green) appear dispersed in different directions and weakly connected. Direct gene-to-isoform alignment is ill-posed in this setting. (Right) Conditioning on tissue context (Brain, Liver, Muscle) resolves this ambiguity by splitting the gene into tissue-specific representations, which form tight clusters with their corresponding isoforms. Solid edges indicate strong, context-aware alignments. The gradient arrow illustrates the transition from an unconditioned to a context-resolved configuration.

## F   ILLUSTRATIONS FOR CROSS-OMICS ALIGNMENT

Due to alternative splicing in eukaryotes and gene reuse in prokaryotes, the relationship between DNA and proteins is one-to-many. As shown in Figure 10 (Left), a gene can produce multiple isoforms whose expression levels vary across tissues, meaning the same DNA sequence may serve different functions in different contexts. An ideal bridging strategy, as illustrated in Figure 10 (Right), should align DNA to different context-specific centers, enabling tissue-aware interpretation.

## G   USE OF LLM

We use a large language model (LLM) for minor edits to grammar, phrasing, and readability. The LLM is not involved in designing the method, developing theoretical results, or conducting experiments. All technical contributions, equations, and results are solely the work of the authors.

