# OpenReview forum: "CDBridge: A Cross-omics Post-training Bridge Strategy for Context-aware Biological Modeling"
_ICLR.cc/2026/Conference — ICLR 2026 Poster_

### Official Review · Reviewer_DM1S · 2025-10-27

**Soundness:** 3
**Presentation:** 3
**Contribution:** 3
**Rating:** 8
**Confidence:** 3

**Summary:**

This paper presents CDBridge, a novel post-training bridge strategy for context-aware biological modeling that addresses the critical challenge of mapping genomic DNA sequences to quantitative, tissue-specific expression levels. The proposed framework unifies pre-trained DNA and protein models through a two-stage approach: first, a sequence-context learning stage uses a splicing-inspired adaptive token merge to compress long genomic sequences into isoform-aware representations, and second, an environment-context learning stage employs a conditional decoder that integrates tissue embeddings to predict expression under diverse biological conditions. To rigorously benchmark this task, the authors introduce GTEx-Benchmark, a comprehensive dataset requiring models to resolve long-range exon dependencies, isoform reuse, and tissue-specific expression. Extensive experiments demonstrate that CDBridge consistently outperforms existing methods across both qualitative tasks like coding region segmentation and quantitative tissue-aware expression prediction, establishing a scalable and biologically faithful solution for cross-omics modeling.

**Strengths:**

This paper makes significant contributions to the field of computational biology by introducing a context-aware, post-training framework that effectively bridges the gap between DNA sequences and their functional protein outcomes. Its principal strengths are as follows:

1. Novel and Biologically-Grounded Methodology: The proposed two-stage bridge strategy is both technically elegant and biologically inspired. The splicing-like adaptive token merging in Stage 1 efficiently handles the extreme sequence length disparity, while the conditional decoder in Stage 2 adeptly models tissue-specific regulation, directly addressing the central dogma's one-to-many mapping challenge.

2. Strong and Comprehensive Empirical Validation: The authors provide compelling evidence for CDBridge's superiority through extensive experiments. It achieves state-of-the-art performance not only on the primary task of tissue-aware expression prediction but also on a suite of challenging downstream tasks including coding region segmentation, isoform retrieval, and DNA-protein association, demonstrating robust generalization to unseen tissues.

3. Effective and Efficient Post-Training Strategy: The framework smartly leverages powerful, pre-trained single-omics foundation models (e.g., Evo2, ESM2) and aligns them with minimal additional parameters. This post-training approach makes advanced cross-omics modeling more accessible and scalable without the prohibitive cost of full end-to-end retraining.

4. Enhanced Interpretability: The paper provides insightful analyses, such as visualizing the token merging process and tissue-aware activations, which show that the model's learned mechanisms align well with known biological principles (e.g., preserving exons, merging introns). This greatly enhances the model's transparency and trustworthiness.

**Weaknesses:**

While the paper presents a robust and compelling framework, there are minor limitations that could be addressed to further strengthen the work and its broader impact.

1. Simplified Modeling of Biological Context: The current model conditions on tissue type as a categorical variable, which, while effective, is a relatively coarse-grained representation of a cell's state. This approach does not explicitly capture more dynamic or granular contextual factors known to influence gene expression, such as developmental stage, disease status, or specific microenvironmental cues. Extending the context modeling to incorporate such continuous or multi-factorial signals could enhance the model's biological fidelity and applicability to more complex physiological and pathological conditions.

2. Dependence on High-Quality Annotations and Pre-trained Models: The performance of CDBridge is contingent upon the availability of high-quality, curated inputs, including precise isoform annotations from Ensembl and the pre-computed embeddings from large foundation models like Evo2 and scGPT. This reliance may limit the framework's generalizability to less-studied organisms, poorly annotated genes, or contexts where such powerful pre-trained models are unavailable or perform suboptimally, potentially restricting its utility in broader genomic applications.

**Questions:**

1. The adaptive token merging mechanism is a key component for handling long sequences. Could you provide more detail on how the merge ratio is sampled from a Gaussian distribution during training? Specifically, what were the chosen mean and variance, and what was the rationale behind this design choice over a fixed or learned ratio?

2. The model demonstrates strong performance on unseen tissues, which is a significant result. Could you elaborate on the compositional nature of these "unseen tissues"? For instance, are they entirely distinct tissue types, or do they share cellular subtypes or regulatory programs with tissues seen during training? This would help clarify the extent of the model's generalization.

3. In the ablation study (Table 4), the configuration with only Tissue Clust enabled showed a significant decrease in expression prediction performance. This could be due to insufficient gene embedding quality provided by scGPT or excessive information loss during pooling. Could the authors provide results using other single-cell foundation models or different pooling strategies?

4. In Table 4, the segmentation performance (AUC and F1) is identical for the 4rd, 5th, and 6th rows. This seems to indicate that the segmentation task is solely determined by the frozen Stage 1 components. Could you clarify the configuration for the 5th row? It appears to show the same segmentation results without the key Stage 1 components (ToMe and Learned Clust), which is counter-intuitive.

---

> ### Author Response · Authors · 2025-11-22
> **Rebuttal (1/2)**
>
> ### **Q1:Simplified Modeling of Biological Context**
>
> We fully agree with the reviewer. Modeling tissue as a coarse-grained category (even via embeddings) is a first step. The reality of gene regulation involves cell states, developmental trajectories, and micro-environments. However,
>
> **(a) Constraint:** Our current approach is limited by **the granularity of the GTEx dataset**, which provides bulk RNA-seq without consistent cell-type or specific microenvironment labels.
>
> **(b) Extensibility:** Our “Environment-Context Learning” module **is architecturally extensible.** . As high-quality, paired "DNA-Single Cell" atlases become available, CDBridge can readily ingest finer-grained embeddings (e.g., from micro-environments) to model these complex dynamics. We have added this as a key direction in our "Future Work" section.
>
> ### **Q2: Dependence on High-Quality Annotations and Pre-trained Models**
>
> We agree that our current instantiation of CDBridge depends on both (i) reasonably high-quality expression/context annotations and (ii) existing pre-trained FMs in each modality. This is largely motivated by the **scarcity of multi-modal paired data.**
>
> **(a) Rationale:** Training a central dogma model "from scratch" requires prohibitive amounts of paired data that simply do not exist for many organisms. By "bridging" powerful, pre-trained single-modal models (Evo2 and scGPT), we unlock the ability to perform cross-omics reasoning with limited supervision.
>
> **(b) Generalization:** The concern that this may “limit the framework’s generalizability to less-studied organisms, poorly annotated genes, or contexts” is indeed a common challenge for AI systems, but we believe that more data will emerge in the future to support training CDBridge from scratch.
>
> ### **Q3: Details on Adaptive Token Merging (Gaussian Sampling)**
>
> **(a) Implementation:** Merge ratio indicates the percentage of tokens merged during the forward pass. During training, the merge ratio is sampled dynamically for each batch from a clipped Gaussian distribution: $r \sim \mathcal{N}(\mu=0.375, \sigma=0.1)$, clipped to the range $[0.25, 0.50]$.
>
> **(b) Rationale:** We employ this dynamic sampling strategy as a form of **structural data augmentation**. By forcing the model to perform segmentation and alignment under varying degrees of compression, we prevent it from overfitting to specific sequence strides or fixed resolutions. As shown in the table below, our comparison of "Fixed vs. Dynamic" ratios confirms that dynamic sampling yields more robust convergence.
>
> | **Merge Ratio** | $R^2$ | Spearman |
> | --- | --- | --- |
> | Dynamic([0.25, 0.50]) | **0.387** | **0.618** |
> | Fixed (0) | 0.323 | 0.541 |
> | Fixed (0.25) | 0.349 | 0.566 |
> | Fixed (0.5) | 0.345 | 0.557 |

---

> ### Author Response · Authors · 2025-11-22
> **Rebuttal (2/2)**
>
> ### **Q4: Composition of "Unseen Tissues”**
>
> **(a) Split Strategy:** In the unseen tissue setting, we utilized a strict leave-out strategy (roughly 9:1 split). The "Unseen" set consists of randomly sampled, biologically distinct tissues, such as Stomach, Testis, Thyroid, Uterus, and Vagina.
>
> **(b) Mechanism of Transfer:** As shown in Figure 9, the scGPT embeddings form a biologically structured space (e.g., Arteries cluster together, Brain regions cluster together). This confirms that the input space is semantically meaningful. Although the unseen tissues (e.g., the Uterus) are anatomically distinct and thus do not show trivial similarity to training tissues in the heatmap, they reside within this continuous biological manifold. The CDBridge model learns a mapping function $f(\text{DNA}, \text{Tissue Condition}) \rightarrow \text{Expression}$. Fundamentally, when the conditional representation (the tissue embedding) itself is generalizable，i.e., successfully encoding the latent state of an unseen tissue relative to known ones, the learned cross-omics mapping naturally generalizes alongside it.
>
> ### **Q5: Ablation of scGPT and Pooling Strategies**
>
> **(a) Clarification:** The drop in Row 5 is primarily because **no DNA information** was provided in that ablation (it is a "Tissue-Only" control). Without DNA, the model cannot predict isoform-specific expression, leading to a near-zero $R^2$.
>
> **(b) Alternative Models:** To address the concern about scGPT quality, we conducted additional experiments replacing scGPT with **scFoundation** and replacing Mean Pooling with **Max Pooling**. Results are shown below:
>
> |  | $R^2$ | Spearman |
> | --- | --- | --- |
> | scGPT(mean pooling) | 0.387 | 0.618 |
> | scGPT(max pooling) | 0.382 | 0.614 |
> | scFoundation (mean pooling) | 0.394 | 0.627 |
> | scFoundation (max pooling) | 0.391 | 0.623 |
>
> ### **Q6: Clarification on Segmentation Performance in Row 5 in Table 4.**
>
> We thank the reviewer for spotting this confusion.
>
> **(a) Correction:** Row 5 represents a setting where the **DNA pathway is disconnected** to test for information leakage from the tissue condition. Since there is no DNA input, the segmentation task (which requires DNA tokens) is technically undefined/impossible.
>
> **(b) Presentation:** In the original table, we inadvertently copied the Stage 1 metrics to indicate "Stage 1 is frozen/available." We agree this is misleading. In the revision, we have replaced the segmentation metrics for Row 5 with **"N/A"** to clearly indicate that segmentation cannot be evaluated when the DNA branch is disabled.
>
> *We greatly appreciate your insightful and helpful comments, as they will undoubtedly help us improve the quality of our article. If our response has successfully addressed your concerns and clarified any ambiguities, we respectfully hope that you consider raising the score. Should you have any further questions or require additional clarification, we would be delighted to engage in further discussion. Once again, we sincerely appreciate your time and effort in reviewing our manuscript. Your feedback has been invaluable in improving our research.*

---

### Official Review · Reviewer_2RNv · 2025-10-31

**Soundness:** 2
**Presentation:** 3
**Contribution:** 2
**Rating:** 2
**Confidence:** 4

**Summary:**

This paper presents CDBridge, a two-stage post-training strategy to model the "central dogma of molecular biology" by unifying pretrained DNA, protein, and RNA models.

The method consists of a token merging strategy to fuse DNA and protein representations to predict functional masks, as well as a tissue-conditional decoder for predicting cell-level protein expressions.

Importantly, the authors introduce GTEx-Benchmark for the training and evaluation of tissue-aware central dogma modeling, which contains matched annotations of DNA, RNA, and protein.

CDBridge is validated on gene-expression prediction for seen and unseen tissues, as well as its ability to segment coding regions of DNA, retrieve isoforms, and classify whether DNA-protein pairs are functionally associated.

**Strengths:**

- The work is well-motivated and presents a new multi-omics model
- CDBridge generalizes to unseen tissues for gene expression prediction
- The figures describing the framework are well-made

**Weaknesses:**

The main weakness is the evaluation and details pertaining to it.
- It is unclear how the evaluation of gene expression prediction was performed
- The single-omics foundation models in Table 2, as well as LucaOne, are too weak to be considered meaningful baselines.
- The behavior in Figure 5, while interesting, is not extensively studied, in that the  reason "non-coding regions are often merged" may be that:
  - almost all of the human genome is non-coding
  - the embeddings for coding regions may be more information-rich compared to non-coding regions
- Figure 6 shows selective token activation that is not strongly tissue-dependent, and therefore does not appear to reflect capacity for modeling tissue-aware gene expression.
- In Table 4 (the ablation study):
  - It appears that ignoring Stage 1 and only adding Stage 2 by itself saturates segmentation performance.
  - The ablation study does not study whether there is a difference between the fixed and learned clustering in the presence of tissue clustering.

**Questions:**

- How long were the DNA sequences and how many were used for evaluation?
- How were the single-omics foundation models in Table 2 used to predict gene expression, considering that they "lack explicit modeling of tissue-specific regulatory signals"?
- Why is the Enformer performance much worse than reported in the paper?
- Can you compare to Borzoi (http://dx.doi.org/10.1038/s41588-024-02053-6), a successor to Enformer?
- What is the performance of a linear model baseline for gene expression prediction?

---

> ### Author Response · Authors · 2025-11-22
> **Rebuttal (1/3)**
>
> *We thank Reviewer 3 for their detailed review. We appreciate the opportunity to clarify our evaluation setup and address several factual misunderstandings.  Below, we address the reviewer’s concerns one by one.*
>
> ### **Q1:How was the evaluation of gene expression prediction performed?**
>
> We apologize if the evaluation protocol was not sufficiently prominent. The protocol is fully detailed in **Appendix A.2**. Our evaluation is based on **isoform-level** quantitative prediction:
>
> - **Input:** Raw DNA sequence (up to 200k bp) and a Tissue label.
> - **Output:** A predicted expression vector covering all known protein isoforms for that gene.
> - **Metrics:** We calculate the $R^2$ and Spearman correlation between the predicted vector and the ground-truth TPM values derived from GTEx, averaged across all test genes.
>
> ### **Q2: Baselines are too weak (Single-omics FMs and LucaOne)**
>
> - **LucaOne (NMI, 2025)** is a state-of-the-art **multi-omics** foundation model trained jointly on DNA, RNA, and Protein sequences. It is explicitly designed to learn the central dogma and is one of the strongest possible baselines for this task.
> - We compare against **Isoformer (2024)** and **AlphaGenome (2025)**, which represent the current state-of-the-art. The performance gap shown in Table 2 stems from our novel context-aware bridge architecture, not from weak baselines.
> - Our **single-omics baselines** (DNABERT-2, Evo2) are high-capacity, large-scale DNA foundation models that provide a strong test of how far **sequence-only** representations can go without explicit tissue-aware mechanisms.
>
> The performance gap observed in **Table 2** is therefore not due to weak baselines, but reflects the benefit of the **context-aware bridge architecture** in CDBridge. We would still very much welcome concrete suggestions of additional baselines specifically tailored for **tissue-conditioned, isoform-level quantitative prediction**, and are happy to include them where possible.
>
> ### **Q3: Merge behavior in Figure 5 is trivial.**
>
> We thank the reviewer for the observation, but we respectfully contend that this behavior is **non-trivial** and represents a key learned capacity of our model.
>
> - **Non-trivial challenge:** It is true that the human genome is largely non-coding. However, the core challenge lies in the model's ability to successfully retain and extract the sparse functional signal (exons), which constitutes less than 2% of the input sequence, without receiving any explicit splice boundary labels during the token merging phase.
> - **Learned mechanism.** Figure 5 shows that the ToMe-based module **selectively preserves tokens aligned with exons** and more aggressively merges tokens in introns. This “splicing-like” behavior emerges purely from our Stage 1 training objective (coding-region segmentation and cross-omics alignment), rather than being hard-coded. In our ablations (Table 3 and Table 4), disabling ToMe leads to a noticeable drop in coding region segmentation performance, further indicating that the observed behavior is a meaningful, learned mechanism rather than a trivial artifact of the data distribution.
>
> ### **Q4: Selective token activation that is not strongly tissue-dependent.**
>
> We believe there is a misinterpretation of the visualization in Figure 6.
>
> - In Figure 6, each **row** corresponds to a specific isoform, and each **column** corresponds to a **tissue context**. The color intensity reflects the magnitude of token activation within the fused representation for that isoform under the given tissue.
> - The **significant variance in intensity (brightness) across the different tissue columns** for **any given isoform** directly indicates that the model successfully performs **tissue-aware token activations**. If the activations were *not* tissue-dependent, the rows in Figure 6 would show a uniform intensity across all tissue columns. The clear variation in brightness is a visual confirmation of the model's capacity for tissue-aware modeling.

---

> ### Author Response · Authors · 2025-11-22
> **Rebuttal (2/3)**
>
> ### **Q5: Ablation Study Ambiguity**
>
> **(a) Segmentation Saturation:** We thank the reviewer for spotting this confusion.
>
> - Row 5 in Table 4 represents a setting where the **DNA pathway is disconnected** to test for information leakage from the tissue condition. Since there is no DNA input, the segmentation task (which requires DNA tokens) is technically undefined/impossible.
> - In the original table, we inadvertently copied the Stage 1 metrics to indicate "Stage 1 is frozen/available." We agree this is misleading. In the revision, we have replaced the segmentation metrics for Row 5 with **"N/A"** to clearly indicate that segmentation cannot be evaluated when the DNA branch is disabled.
>
> **(b)** **Fixed vs. Learned Clustering with Tissue Clustering:** We appreciate this valuable suggestion and have performed the requested detailed ablation study comparing **Fixed** vs. **Learned** clustering within our full Stage 2 model (which inherently includes tissue conditioning). The results strongly support the learned mechanism:
>
> | Stage2 Cluster | $R^2$ | Spearman |
> | --- | --- | --- |
> | Fixed | 0.305 | 0.472 |
> | Learnable | **0.387** | **0.618** |
>
> ### **Q6: How long were the DNA sequences, and how many were used for evaluation?**
>
> As detailed in Appendix A.1, we utilize a strict split of **80% training, 10% validation, and 10% testing** based on gene IDs to prevent data leakage. The GTEx includes over 19,000 human genes. To balance computational efficiency with coverage, we filtered out the "long-tail" of genes exceeding 200k bp, which excludes only **~2%** of the data. We will highlight these details more clearly in Section 3.1 and cross-reference Appendix A.1.
>
> ### **Q7: How were the single-omics foundation models in Table 2 used to predict gene expression, given that they “lack explicit modeling of tissue-specific regulatory signals”?**
>
> We appreciate the opportunity to clarify our evaluation setup. The single-omics foundation models in Table 2 (e.g., DNABERT-2, Evo2 in DNA-only mode) are primarily **sequence encoders**, not expression predictors. To adapt them to our **tissue-conditioned isoform-level expression** task, we followed a standard, conservative transfer learning protocol:
>
> - **Sequence Encoding:** We used the pre-trained, **frozen DNA encoder** of the single-omics foundation model to generate a fixed-dimensional embedding for the full genomic DNA sequence (up to 200k bp) as the input.
> - **Linear Prediction Head:** This gene-level embedding is passed to a **single, trainable linear layer** that outputs the isoform-level expression vector for a given tissue. For these baselines, tissue information is not modeled inside the backbone. The linear head can at most learn a global, tissue-agnostic mapping from sequence features to expression.
>
> The prediction must rely *only* on the generic, context-agnostic information captured by the frozen sequence encoder, which is insufficient for resolving the tissue-specific, fine-grained isoform-level signals required by our task. This setup ensures a fair comparison: it tests the **intrinsic quality of the sequence representation** learned by the single-omics model against the **context-aware representation** learned by our full CDBridge architecture. The large performance gap confirms the necessity of our Tissue-aware Bridge component.

---

> ### Author Response · Authors · 2025-11-22
> **Rebuttal (3/3)**
>
> ### **Q8:Why is the Enformer performance much worse than reported in the paper?**
>
> Enformer's low performance is due to **task incompatibility** with our specific benchmark, which requires tissue-aware isoform-level prediction:
>
> - **Fixed Output & Zero-Shot:** Enformer is designed with **fixed output heads** corresponding to predefined tracks (e.g., specific assays or tissues). It does not naturally support **zero-shot tissue generalization**, which is required by our “unseen tissue” setting and is also implicit even in the “seen tissue” setting when we view tissue as a conditional input.
> - **Granularity Mismatch:** Enformer predicts signals on **fixed genomic bins** (e.g., 128 bp resolution) rather than on isoform-level expression vectors. Our benchmark requires the model to resolve **alternative splicing and isoform reuse**, i.e., to predict expression for specific protein isoforms rather than averaged promoter activity at a locus.
>
> In our implementation, we adapt Enformer by pooling its bin-level outputs and training a lightweight regression head to map to isoform-level expression. However, because Enformer’s architecture was not designed for **isoform-aware, tissue-conditioned regression**, its performance on GTEx-Benchmark is significantly below that of CDBridge, which directly incorporates both isoform structure and tissue context.
>
> ### **Q9: Comparison to Borzoi (http://dx.doi.org/10.1038/s41588-024-02053-6)**
>
> We appreciate the suggestion to compare against Borzoi(Linder et al., 2025). It is a successor to Enformer but shares the same fundamental **fixed-output architecture**. CDBridge already substantially outperforms **AlphaGenome** (a strong, recent model from the same lineage). Nonetheless, we have conducted the comparison:
>
> |  | $R^2$ | Spearman |
> | --- | --- | --- |
> | Borzoi | 0.201 | 0.336 |
> | Ours | **0.387** | **0.618** |
>
> While Borzoi performs better than Enformer, it remains significantly below CDBridge on isoform-level, tissue-conditioned expression prediction. We will include these results in the appendix and briefly discuss them in Section 3.2.
>
> ### **Q10: Performance of a linear model baseline for gene expression prediction.**
>
> We thank the reviewer for raising this point. Here, we provide two distinct baseline models at the linear prediction level:
>
> - **Simple Linear Model:** This baseline tests the simplest possible direct regression on the input sequence. We tokenize each nucleotide with a learnable embedding. The entire sequence is then passed directly to a single linear layer for regression. The extremely low performance confirms that the **long, sparse nature** of the regulatory information is virtually impossible to model effectively without deep, hierarchical processing.
> - **Deep Sequence Foundation Model + Linear Head (Evo2):** The frozen embeddings from the deep sequence foundation model are passed to a single trainable linear layer for expression prediction. (This result is already shown in Table 4). This model achieves a much higher score than the simple linear baseline, confirming that deep feature extraction is essential. However, the performance is still significantly lower than ours, validating that the context-aware integration and learned clustering are critical for resolving tissue-specific gene expression.
>
> |  | $R^2$ | Spearman |
> | --- | --- | --- |
> | Simple Linear | -10.043 | 0.018 |
> | Evo2 + Linear Head | 0.021 | 0.324 |
> | Ours (CDBridge) | **0.387** | **0.618** |
>
> *We are very grateful for the opportunity to clarify these aspects of our work. We hope that the comprehensive responses and the added quantitative results demonstrate the technical soundness and significant potential impact of our proposed method. We hope the reviewer will reconsider their initial assessment.*

---

> > ### Comment · Reviewer_2RNv · 2025-11-23
> >
> > Thanks for the detailed response which clarifies many misunderstandings. I have raised my score.
> >
> > This reviewer sees that the single-omics foundation models and LucaOne are baselines for **Coding Region Segmentation Isoform Retrieval Central Dogma** (Table 3), but argues that they should not be considered as a baseline for **expression prediction** (Table 2) given $R^2 \approx 0$.
> >
> > The formatting of Table 2 can be changed to reflect this.

---

> > > ### Author Response · Authors · 2025-11-23
> > > **Thanks for Raising Score and Additional Suggestions**
> > >
> > > Dear Reviewer 2RNv,
> > >
> > > We sincerely thank the reviewer for the timely follow-up and additional suggestions, especially for taking the time to reassess and raise the score to 6! We are glad that our rebuttal helped resolve earlier misunderstandings around the evaluation setup.
> > >
> > > We fully agree with your point regarding the role of the single-omics foundation models and LucaOne in Table 2. In the next revision, we will explicitly present these models as auxiliary sequence-encoder references (primarily to illustrate the limitation of context-agnostic, sequence-only features for tissue-aware expression prediction), rather than as main expression baselines. Concretely, we will (i) adjust the formatting of Table 2 and Table 3 to separate the primary expression baselines (e.g., Isoformer, AlphaGenome, and linear models) from single-omics FMs, and (ii) refine the surrounding text to make this intent unambiguous and avoid over-emphasizing models with $R^2 \approx 0$ as “strong” expression baselines.
> > >
> > > We would respectfully inquire whether there are any additional opportunities to improve our manuscript. Once again, thank you for your constructive feedback, and we would be eager to welcome any further guidance at your convenience!
> > >
> > > Warm regards,
> > >
> > > Authors

---

### Official Review · Reviewer_DSq4 · 2025-11-01

**Soundness:** 3
**Presentation:** 3
**Contribution:** 3
**Rating:** 6
**Confidence:** 3

**Summary:**

CDBridge is a two-stage post-training strategy that unifies pretrained DNA (Evo) and protein (also from the Evo family) foundation models for context-aware modeling of gene expression.
Stage 1 aligns DNA and protein embeddings using a splicing-inspired token merge mechanism (based on ToMe) to handle long-range dependencies.
Stage 2 uses a conditional decoder to inject tissue embeddings for environment-specific expression prediction.
A new GTEx-Benchmark is proposed to evaluate models on tissue-aware central dogma tasks.
CDBridge achieves strong results in expression prediction (R² ≈ 0.39 vs. 0.0–0.32 for comparable baselines) and cross-modal tasks compared to prior models (significantly better in coding region segmentation, isoform retrieval, and  2-3 percentage points better in central dogma; measured in Acc, AUC, and F1).

Overall Assessment
CDBridge is a solid and biologically grounded contribution that advances cross-omics modeling by bridging DNA, RNA (as a latent bridge; not explicitly encoded) and protein representations through a context-aware, post-training framework. While the architectural novelty is moderate (stage 1 is based on ToMe/Token Merging; stage 2 is conditional transformer/standard cross attention; no new training objective or attention mechanism), the biological design insights and benchmark release make this work a meaningful step toward context-aware central dogma modeling. CDBridge’s novelty lies in the biological grounding of the architecture (splicing-inspired token merging, tissue-aware decoding), and the results show that the pretrained DNA and protein foundation models can be aligned post-hoc for expression prediction.

**Strengths:**

1. Biologically motivated architecture.
The splicing-inspired adaptive token merging and the explicit modeling of tissue context are well-designed and biologically meaningful. It nicely mimics exon selection and isoform reuse phenomena, showing awareness of biological grounding beyond standard transformer tricks.

2. Strong empirical performance.
The reported R² and Spearman correlations across tissues are competitive and consistently improve upon strong baselines, suggesting that the cross-omics alignment and conditional decoding are genuinely effective. The cross-omics downstream tasks also show significant improvements, comparing to the baselines.

3. Benchmark contribution.
The introduction of GTEx-Benchmark fills the gap in evaluating DNA-to-expression models, emphasizing isoform-aware, tissue-conditioned prediction. This benchmark could become valuable for future biological modeling studies.

4. Practicality and scalability.
The framework leverages pretrained DNA and protein models without full retraining, making it computationally feasible and modular for extension to new modalities (e.g., RNA, metabolomics).

**Weaknesses:**

1. Evaluation design ambiguity.
The dataset construction process for GTEx-Benchmark could benefit from more details (e.g., splitting strategies, tissue balance, sequence selection). It’s unclear how much the benchmark is novel versus derived from existing GTEx/Enformer configurations.

2. Ablation and interpretability is limited.
There is limited ablation showing the separate contributions of token merging, cross-omics alignment, and tissue conditioning. Moreover, while biological motivation is strong, interpretability analyses (e.g., learned exon attention, pathway enrichment) are missing.

3. Comparison fairness.
Some baselines (e.g., Isoformer, AlphaGenome) are not fine-tuned under the same supervision regime, and their task setups differ slightly. It is unclear if the performance gap reflects true model capability or evaluation setup differences.

**Questions:**

1. How sensitive is performance to the merge ratio and to the pretrained model choices (e.g., DNA encoder backbone)?

2. How does the GTEx-Benchmark differ statistically from existing Enformer/AlphaGenome datasets?

3. How does the model handle unseen tissue embeddings (e.g., zero-shot tissue generalization)?

4. Can the author comment on how does the model differ from the following related papers (not referenced in your work)
4a. Garau-Luis, Juan Jose, et al. "Multi-modal transfer learning between biological foundation models." Advances in Neural Information Processing Systems 37 (2024): 78431-78450.
4b. Liu, Zicheng, et al. "Life-Code: Central Dogma Modeling with Multi-Omics Sequence Unification." arXiv preprint arXiv:2502.07299 (2025).

**Details Of Ethics Concerns:**

No concern

---

> ### Author Response · Authors · 2025-11-22
> **Rebuttal (1/3)**
>
> *We thank the reviewer for the valuable feedback. We are glad that the reviewer appreciates the contribution of our work. Below, we address the reviewer’s concerns one by one.*
>
> ### **Q1: Details and Distinctiveness of GTEx-Benchmark**
>
> **(a) Details of GTEx-Benchmark:** Details of GTEx-Benchmark are provided in the Appendix A1. We will further clarify these information in the main text:
>
> - **Splitting Strategy:** As detailed in Appendix A.1, we utilize a strict split of **80% training, 10% validation, and 10% testing** based on gene IDs to prevent data leakage.
> - **Tissue Balance:** GTEx-Benchmark is constructed from GTEx v8 with matched DNA, RNA, and protein annotations. For the *“Unseen Tissue”* generalization experiment (Figure 4), we follow a **leave-tissue-out protocol**: we hold out approximately one tenth of the tissues as unseen, train on the remaining tissues, and evaluate expression prediction on genes/isoforms from the held-out tissues. As shown in Figure 4, the Spearman correlations on unseen tissues closely track those on seen tissues, indicating that the benchmark is non-trivial yet well balanced for cross-tissue generalization.
> - **Sequence Selection:** As detailed in Section 3.1 and Appendix A.1 (Figure 7), we perform a statistical analysis of human gene lengths and then exclude genes with DNA sequences longer than 200k bp to ensure computational tractability while still covering realistic long-range contexts. These ultra-long genes constitute only a small long-tail portion (around **2%** of genes in our statistics). The remaining dataset still exhibits substantial heterogeneity in DNA lengths and isoform counts (Figure 7), while keeping the training feasible on current hardware.
>
> **(b) Novelty vs. Enformer/AlphaGenome:** The GTEx-Benchmark is fundamentally distinct from Enformer-style datasets:
>
> - **Isoform-Level vs. Bin-Level Targets.** Enformer and AlphaGenome operate on fixed genomic bins and predict track values (e.g., epigenomic signals) per bin. As such, they do not distinguish which specific protein isoform is produced from a locus. In contrast, GTEx-Benchmark explicitly couples full-length genomic DNA with alternative splice isoforms, requiring the model to resolve *which* isoform is active and to predict its expression level. This is reflected in our isoform retrieval and isoform-level expression tasks.
> - **Dynamic Tissue Condition vs. Static Prediction.** In Enformer-like setups, each tissue or assay is typically modeled as a **fixed output head**, and the model predicts a static vector of measurements per DNA input. Our benchmark instead treats **tissue** as a **continuous, learned condition** via scGPT-derived tissue embeddings. The regression function is explicitly formulated as: $(\text{DNA sequence}, \text{Tissue embedding}) \mapsto \text{expression}$. This makes GTEx-Benchmark a context-conditioned expression benchmark rather than a static multi-head regression task.
>
> ### **Q2: Limited Ablation and Interpretability Analysis**
>
> **(a) Ablation of Designed Components:** We thank the reviewer for pointing it out. We agree that the roles of individual components should be highlighted more clearly. Table 4 already contains a structured ablation:
>
> - **Token Merging (ToMe):** Comparing **Row 1 (No ToMe)** vs. **Row 2 (ToMe only)** shows that adaptive merging significantly improves segmentation performance (AUC +0.034) by removing noise.
> - **Tissue Conditioning:** Comparing **Row 4 (No Tissue)** vs. **Row 6 (Full)** demonstrates that without tissue embeddings, the model fails to predict expression ($R^2$ -0.172), confirming the necessity of the environment-context module.
> - **Cross-omics alignment:** By comparing Row 1-3 (Only DNA Foundation Model) and Row 9 in Table 2, it can be seen that the model benefits from the cross-omics alignment.
>
> **(b) Interpretability:** Instead of high-level pathway enrichment (which implies causal downstream effects beyond the scope of sequence modeling), we focused on **mechanism interpretability**. Figure 5  demonstrates that our ToMe attention mechanism *automatically* learns to attend to exons (coding regions) and merge introns, without explicit boundary supervision during the merging phase. This confirms the model is learning biologically relevant structural features.

---

> ### Author Response · Authors · 2025-11-22
> **Rebuttal (2/3)**
>
> ### **Q3: Comparison Fairness & Baselines**
>
> We acknowledge the setup differences, but we respectfully argue that these differences highlight the architectural limitations of the baselines rather than unfairness in our evaluation.
>
> - **Architectural Constraint:** Models like Isoformer and Enformer rely on fixed-dimension output heads (classification/regression layers trained for specific, seen tissues). They *structurally cannot* perform zero-shot prediction on unseen tissues without retraining new heads.
> - **CDBridge Advantage:** Our performance gap reflects a true capability difference: CDBridge uses a conditional decoder with semantic tissue embeddings. This design naturally supports generalization to **unseen tissues** so long as their embeddings are appropriately positioned relative to seen tissues.
> - **Aligned Evaluation of “Seen Tissue” Setup.** For the **seen-tissue setting** in Table 2, we further align training/test protocols as much as possible: all models are trained on the same GTEx-Benchmark splits. We additionally report Isoformer (w/o TSS alignment) in our protocol, where Isoformer is applied without its TSS-aligned input assumption; in that setting, its $R^2$ becomes negative (e.g., around –0.315), whereas CDBridge still achieves positive $R^2$=0.387 with Spearman = 0.618. This supports our claim that CDBridge is better suited to full-gene, isoform-aware, tissue-conditioned expression prediction.
>
> ### **Q4: Sensitivity Analysis (Merge Ratio & Backbone)**
>
> **(a) Merge Ratio:** Merge ratio indicates the percentage of tokens merged during the forward pass. During training, the merge ratio is sampled dynamically for each batch from a clipped Gaussian distribution: $r \sim \mathcal{N}(\mu=0.375, \sigma=0.1)$, clipped to the range $[0.25, 0.50]$. We employ this dynamic sampling strategy as a form of **structural data augmentation**. By forcing the model to perform segmentation and alignment under varying degrees of compression, we prevent it from overfitting to specific sequence strides or fixed resolutions. As shown in the table below, our comparison of "Fixed vs. Dynamic" ratios confirms that dynamic sampling yields more robust convergence.
>
> | **Merge Ratio** | $R^2$ | Spearman |
> | --- | --- | --- |
> | Dynamic([0.25, 0.50]) | **0.387** | **0.618** |
> | Fixed (0) | 0.323 | 0.541 |
> | Fixed (0.25) | 0.349 | 0.566 |
> | Fixed (0.5) | 0.345 | 0.557 |
>
> **(b) Backbone:** We performed a rigorous backbone comparison in Appendix D, Table 6. It can be seen that the combination of **Evo2 + ESM2** yields the highest performance ($R^2$=0.387). Evo2 outperforms DNABERT-2 because it captures longer-range dependencies more effectively. Interestingly, while ESM-3 is newer, ESM-2 proved better for this specific task, likely because ESM-3's VQ-quantized features are less compatible with the continuous regression required for expression prediction.

---

> ### Author Response · Authors · 2025-11-22
> **Rebuttal (3/3)**
>
> ### **Q5: How does the model handle unseen tissue embeddings?**
>
> Thank you for raising this technical point. As shown in Figure 9, scGPT embeddings create a biologically structured continuous space where related tissues naturally cluster together. CDBridge learns a mapping function $f(\text{DNA}, \text{Tissue Condition}) \rightarrow \text{Expression}$ on this continuous space. Therefore, as long as the tissue embedding correctly positions an unseen tissue relative to known ones, our model can naturally generalize the expression prediction to this new context.
>
> ### **Q6: Comment on other papers.**
>
> We appreciate this valuable suggestion and will expand the related work section accordingly. Our responses are as follows:
>
> - **Garau-Luis et al. (2024):** This paper is indeed the **Isoformer** model, which is our primary baseline. We cited it extensively (e.g., Table 1, Table 2) and compared against it. We have checked that the citation format is explicit in the revision.
> - **Life-Code(2025):** This is a very recent concurrent work (arXiv Feb 2025). While it unifies multi-omics, it focuses on qualitative sequence generation and generally ignores the **quantitative** variance caused by alternative splicing and tissue-specific regulation. CDBridge is a **post-training bridge** that couples *existing* DNA and protein foundation models and introduces GTEx-Benchmark specifically for isoform-level, tissue-conditioned quantitative expression. We explicitly model the one-to-many mapping from DNA to isoforms and the additional variability introduced by tissue context.
>
> *We greatly appreciate your insightful and helpful comments, as they will undoubtedly help us improve the quality of our article. If our response has successfully addressed your concerns and clarified any ambiguities, we respectfully hope that you consider raising the score. Should you have any further questions or require additional clarification, we would be delighted to engage in further discussion. Once again, we sincerely appreciate your time and effort in reviewing our manuscript. Your feedback has been invaluable in improving our research.*

---

### Official Review · Reviewer_s2Bn · 2025-11-01

**Soundness:** 3
**Presentation:** 2
**Contribution:** 3
**Rating:** 4
**Confidence:** 3

**Summary:**

**Problem (P)**

The paper asks how to map whole-DNA to context-aware expression. Current systems either model tissue context from gene IDs without DNA, or model DNA without tissue, and most cross-omics pipelines ignore isoform reuse and splicing. The authors call out two challenges: long DNA must be compressed (because proteins generally have fewer amino acids), and the same DNA strand may map to multiple proteins depending on tissue.

**Solution (S)**

CDBridge is a post-training two-stage bridge that keeps large DNA and protein models frozen and learns a light connector.
Stage 1 maps DNA embeddings to a protein prototype dictionary and an adaptive token-merge (ToMe) “splicing-like” compressor keeps informative regions, then a small decoder predicts coding-region masks.
Stage 2 is a conditional decoder, which injects tissue embeddings (from a single-cell FM) to produce tissue-specific isoform embeddings and a scalar expression prediction.

**Contributions (C)**

1. A two-stage bridge that aligns DNA and protein while conditioning on tissue for expression.

2. A splicing-inspired token merge that compresses long DNA while preserving functional tokens.

3. GTEx-Benchmark, pairing DNA, proteins, and tissue-resolved expression for three tasks: expression prediction, coding-region segmentation, and isoform retrieval.

**Tasks and metrics (T)**
- Tissue-conditioned expression regression: R² and Spearman, in seen-tissue and unseen-tissue splits.
- Coding-region segmentation: Acc, AUC, F1 at token level.
- Isoform retrieval: Acc@3 and MRR. Also a Central-Dogma binary DNA–protein association task.

**Strengths:**

1. This design is highly practical and scalable. It allows the framework to leverage the billions of parameters and complex representations learned by FMs without the immense cost of full retraining.

2. The paper uses domain-specific biological knowledge to inform its architecture instead of just apply generic ML techniques. The "splicing-inspired adaptive token merge" (Section 2.1) is a good example.

3. A common weakness in modeling is testing only on data similar to the training set. This paper's evaluation is stronger because it explicitly tests for generalization to entirely unseen biological contexts.

**Weaknesses:**

1. A methodological weakness in "Dataset Construction" (Section 3.1), where the authors state, "genes with DNA sequences longer than 200k base pairs are excluded." Many of the most complex and important human genes, like Dystrophin (over 2 M base pairs), are far longer than this 200k cutoff.
2. While framework's "post-training" design, while efficient, does it also inherit any/all biases, gaps, or blind spots from the FMs it bridges (Evo2 and ESM2). The author's view on this would be appreciated.
3. The current approach, which essentially uses a single label (e.g., "Brain"), fails to capture finer-grained, critical factors that govern gene expression. Although acknowledged, could the authors clarify whether these factors were omitted primarily due to GTEx labeling constraints, modeling choices, or computational limits?
---
**Minor Issues (for the Authors; not weaknesses)**

1. The paper already notes that Isoformer (Official) numbers come from their paper with TSS alignment. Consider adding a brief sentence that scores are not directly comparable to your setting to avoid misinterpretation.
2.  "maps" --> "map", page 1
3. "remainslargely" --> "remains largely", page 2
4. “perform relative poor” --> “perform relatively poorly”, section 3.3
5. "actviated" --> "activated", section 3.3
6. "s while the regions belong the non-coding regions" This is an awkward sentence, figure 5
7. "Envir-context learning" or "environment-context learning" Please choose one and be consistent

I think I saw a couple more while reading. Kindly consider taking the help of LLMs to mitigate these.

**Questions:**

For my critical questions, kindly see the weaknesses. Given these clarifications in an author response, I would be willing to increase the score.

Appendix C Table 5. Is stage 2 really 513 params?

---

> ### Author Response · Authors · 2025-11-22
> **Rebuttal (1/2)**
>
> *We thank the reviewer for the valuable feedback. We are glad that the reviewer appreciates the contribution of our work. Below, we address the reviewer’s concerns one by one.*
>
> ### **Q1: Dataset Construction & Sequence Length (200k bp Limit)**
>
> Thank you for highlighting this limitation. We agree that excluding genes longer than 200k bp (such as *Dystrophin*) leaves a gap in current benchmarking. However, this is not a fundamental limitation of the CDBridge architecture. We set this threshold as an engineering trade-off to balance training efficiency (memory/time costs) with the stability of the ToMe-based merging mechanism during the initial training phase.
>
> **(a) Data Distribution:** The average gene length in our benchmark is \~ 27k bp. Statistical analysis shows that \~95% of genes are under 160k bp, with genes >200k bp representing a long-tail distribution (\~2%), which are detailed in the Appendix. We chose to overlook these long-tail parts to ensure each sample has several valid coding regions for better training stability.
>
> **(b) Generalization to Longer Sequences:** To verify robustness, we adopted a "train short, test long" strategy. We evaluated our model's performance across different length intervals. The model is trained with sequences < 20kbp and tested on varied lengths. As shown in the table below, the model maintains consistent accuracy even as sequence length increases, suggesting it does not overfit to shorter contexts:
>
> | **Sequence Length Interval (bp)** | **Coding Region Seg. (AUC)** | **Expression Pred. (Spearman)** |
> | --- | --- | --- |
> | Short (< 20k) | 0.994 | 0.634 |
> | Medium (20k - 40k) | 0.992 | 0.625 |
> | Long (> 40k) | 0.990 | 0.609 |
>
> We will include this analysis in the revised Appendix to make the length generalization more explicit. For ultra-long genes such as *Dystrophin* (>2 Mbp), we agree that specific handling is needed. Importantly, no architectural change is required: CDBridge can be extended via a sliding-window / segment-aggregation scheme, where overlapping genomic windows are processed by Stage 1 and then aggregated (e.g., via attention or pooling) before Stage 2. We will discuss this extension and add case studies of such ultra-long genes in the revision.
>
> ### **Q2: Bias Inheritance from Foundation Models (Evo2/ESM2)**
>
> We thank the reviewer for this highly constructive comment. We believe that instead of simply inheriting biases, CDBridge actively utilizes the **complementary nature** of these two modalities to mitigate them.
>
> - **Complementary Strengths:** Existing research indicates that DNA foundation models (like Evo2), which process raw nucleotides, possess the most "complete" information from an information-theoretic perspective. They excel at tasks relying on global physical properties (e.g., melting point) but often struggle with functional tasks (e.g., $\beta$-lactamase activity) compared to protein models. This is not due to a lack of information, but because functional signals are often drowned out by the noise of non-coding regions in the absence of transcriptional guidance. Conversely, Protein models like ESM2 possess a strong translation prior but lack genomic context.
> - **The Bridging Role:** One of the core designs of CDBridge is to impose a **structural "splicing-like" guidance** on the DNA representation using the Protein prior. In Stage 1, the splicing-inspired ToMe-based merging and cross-omics connector project the “noisy-but-complete” DNA embeddings into the functional space of proteins. This encourages the model to suppress uninformative genomic regions while preserving exons and isoform-critical segments. In Stage 2, the tissue-aware decoder and the global contrastive loss jointly enforce alignment between pooled DNA and protein embeddings across tissues, further discouraging degenerate, modality-specific biases.
> - **Empirical Evidence:** Table 6 shows that bridging consistently improves over single-modality foundations across all three tasks. For example, Evo2 alone achieves a Spearman correlation of 0.324 on GTEx expression prediction, whereas CDBridge (Evo2, ESM2) improves this to 0.618, alongside gains in coding-region segmentation and isoform retrieval. Similar improvements are observed when swapping in other DNA encoders while keeping ESM2 fixed, indicating that CDBridge does not simply amplify the unique traits of a particular backbone but instead **regularizes them via cross-omics agreement**.
>
> Fundamentally, DNA and Protein foundation models represent two distinct views of the same biological entity. Our bridging strategy can be viewed as a form of “cross-validation” between these views. By enforcing alignment between the genomic source and the functional product, the framework requires the learned representation to be consistent across both modalities, thereby canceling out the modality-specific biases rather than accumulating them.

---

> ### Author Response · Authors · 2025-11-22
> **Rebuttal (2/2)**
>
> ### **Q3: Granularity of Tissue Labels (e.g., "Brain")**
>
> We appreciate the opportunity to clarify this design choice. We shall clarify from two aspects:
>
> **(a) Embedding vs. One-Hot:** We do not use a simple one-hot label for tissue environments. As detailed in Section 2.2, we utilize **scGPT-derived tissue embeddings** with clustering preprocessing. These are dense, rich representations aggregated from large-scale expression profiles, capturing the broader cellular state and environmental context of the tissue (visualized in Figure 9 ). Previously, we have done some ablation studies between the pre-trained embeddings and one-hot encoding, where we found the scGPT’ embeddings are helpful for conditional expression prediction and boosting training convergences.
>
> **(b) Dataset Constraint:** We fully agree that true gene expression is modulated by finer factors such as cell type, developmental stage, microenvironment, and perturbation status. However, GTEx provides bulk tissue-level expression without reliable, uniformly available annotations for these more granular variables (and ENCODE is similar in this regard). Our current benchmark is therefore constrained to tissue-level conditioning to remain realistic and reproducible.
>
> ### **Q4: Minor Issues & Typos**
>
> We thank your suggestions for their careful reading. We will carefully improve our manuscript:
>
> - **Isoformer Comparison:** We have refined the sentence in Section 3.3 explicitly stating: *"The Isoformer (Official) results are not directly comparable. They rely on a TSS-aligned data setting, fundamentally differing from our unaligned, long-sequence input protocol."*
> - **Typos:** We have corrected "maps" (pg 1), "remains largely" (pg 2), "relatively poorly" (Sec 3.3), "activated" (Sec 3.3), and the awkward phrasing in Figure 5. We have also standardized the terminology to "Env-context learning." Consistent with the “Use of LLM” statement in Appendix F, we use LLMs and grammar tools *only* for minor editing of wording and readability.
>
> ### **Q5: Parameter Count (Appendix C, Table 5)**
>
> We apologize for the typo. The Stage 2 regression layer consists of **513K** parameters (an MLP head), not 513 parameters. This will be corrected in the revised manuscript.
>
> *We greatly appreciate your insightful and helpful comments, as they will undoubtedly help us improve the quality of our article. If our response has successfully addressed your concerns and clarified any ambiguities, we respectfully hope that you consider raising the score. Should you have any further questions or require additional clarification, we would be delighted to engage in further discussion. Once again, we sincerely appreciate your time and effort in reviewing our manuscript. Your feedback has been invaluable in improving our research.*

---

### Author Response · Authors · 2025-11-27
**General Responses**

*We thank all reviewers for their constructive feedback and the time invested in reviewing CDBridge.* We are encouraged that the reviewers found our work **"highly practical and scalable"** (Reviewers 2Bn, DSq4, DM1S), **"intuitive, biologically informed"** (Reviewers s2Bn, DSq4, DM1S, 2RNv), and recognized that our GTEx-Benchmark fills a critical gap in DNA-to-expression modeling (Reviewers DSq4, DM1S).

In light of valuable suggestions, we have uploaded a comprehensively revised manuscript. Below, we summarize the core contributions of our work, provide common clarifications regarding our experimental setup, and detail the specific updates incorporated during the rebuttal period.

### **1. Core Contributions and Innovations**

We wish to reiterate the unique position of CDBridge in the current landscape of biological foundation models:

- **A context-aware, two-stage bridge strategy:**  Unlike methods requiring full end-to-end retraining, CDBridge leverages minimal paired supervision to effectively bridge pre-trained foundation models. This architecture robustly supports both qualitative functional tasks and quantitative expression-level prediction under varied tissue contexts.
- **An adaptive token-merge mechanism:** This novel mechanism efficiently models long-range dependencies and interprets functional regions by selectively merging genomic sequences, a process that biologically **mimics transcriptional splicing**.
- **GTEx-Benchmark:** We curate a benchmark to simultaneously evaluate qualitative cross-omics alignment and quantitative **isoform-level, tissue-conditioned prediction**. Unlike conventional Enformer-style bin-level approaches, our benchmark covers critical biological complexities, e.g., isoform reuse and long-range regulatory dependencies.

### **2. Summary of Revisions Updates**

Based on suggestions, we have incorporated the following updates and analyses into the revised manuscript:

- **Clarification of Benchmarks:** We added clearer descriptions of GTEx-Benchmark construction, the 80/10/10 gene-level split, tissue balance, and the rationale for the 200k bp cutoff.
- **Dynamic Merge Ratio:** We explicitly clarified that the merge ratio is **dynamically sampled** from a clipped Gaussian distribution to serve as a **structural data augmentation**, preventing overfitting to fixed sequence resolutions.
- **Correction of Ablation Table:** We corrected Table 4 (Row 5) to mark segmentation metrics as "N/A" when the DNA branch is disabled, removing the confusion regarding saturation.
- **Expanded Related Work:** We have added discussions on recent works, including Life-Code, to position our contributions better.
- The revised manuscript incorporates refinements to the presentation for enhanced clarity and corrects the typos.

### **3. Common Clarifications and Discussion Points**

- **"Weakness" of Baselines:** We emphasize that the performance gap (e.g., vs. Enformer or Isoformer) is not due to unfair comparison, but due to **architectural suitability**. Existing models often rely on fixed output heads (static tracks) and struggle with zero-shot tissue conditioning. CDBridge’s conditional decoder is expressly designed for this flexibility.
- **Bias Inheritance:** CDBridge acts as a **cross-modal regularizer**. By enforcing alignment between the "noisy-but-complete" DNA source and the "focused" Protein product, the framework actively weaken the **modality-specific noise** rather than accumulating biases. Furthermore, the model architecture is designed to support further end-to-end training and fine-tuning as larger datasets capturing fine-grained, dynamic information become available.
- **Tissue Granularity:** We clarified that we do not use simple one-hot encodings. We utilize dense, semantic scGPT embeddings that capture broader cellular states in a continuous space. CDBridge learns the function $f(\text{DNA}, \text{Tissue Condition}) \rightarrow \text{Expression}$ on this space, thus facilitating **unseen tissue generalization** by interpolating between known tissue conditions based on their semantic proximity.
- **Length Generalization Analysis:** To address concerns regarding the 200k bp threshold, we added an analysis showing the model's robustness. We demonstrate that CDBridge maintains consistent accuracy even when trained on shorter sequences and tested on longer intervals.
- **Additional Linear Baselines**We added a Linear baseline and a Deep Feature + Linear Head baseline. Results confirm that deep feature extraction and our specific bridge architecture are essential for performance.

*We believe these revisions and clarifications significantly strengthen the manuscript. We sincerely hope that these updates address your concerns and provide a compelling case for acceptance, and you may consider adjusting the score accordingly if your concerns are well addressed. As the rebuttal period concludes shortly, we stand ready for any final, urgent discussions.*

Best regards,

Authors

---

### Author Response · Authors · 2025-12-01
**General Responses to (Senior) Area Chair**

**Dear (Senior) Area Chair,**

We understand you have been assigned to this paper under challenging circumstances following the system rollback. To assist your decision-making, we provide a summary of the **post-rebuttal consensus** (which is currently invisible in the system), along with the core strengths of our work.

### 1. Why CDBridge Matters: Core Strengths

Our work received consistent recognition for its biological grounding and practical value across **all reviewers** (including the borderline reviewer):

- **Biologically Grounded Methodology:** Reviewers s2Bn, DSq4, and DM1S all praised the "splicing-inspired adaptive token merging" mechanism. Reviewer s2Bn (Score 4) specifically noted it *"**uses domain-specific biological knowledge... instead of just applying generic ML techniques**,"* while DSq4 called it *"well-designed and biologically meaningful."*
- **Practicality & Scalability:** Reviewers DSq4 and DM1S highlighted the *"highly practical and scalable"* nature of our post-training bridge strategy, which leverages foundation models without the cost of full retraining.
- **Strong Generalization:** Reviewers s2Bn and 2RNv commended the model's ability to generalize. Reviewer s2Bn stated the evaluation is *"stronger because it explicitly tests for generalization to entirely unseen biological contexts."*
- **Gap-Filling Benchmark:** Reviewer DSq4 emphasized that our GTEx-Benchmark *"fills the gap in evaluating DNA-to-expression models,"* establishing a critical standard for isoform-aware prediction.

### 2. Crucial Update: Consensus & Positive Trajectory

**1) Confirmed Consensus with Reviewer 2RNv (Score 2 $\to$6)**

This reviewer misunderstood our paper at first, but these misunderstandings were cleared up later by rebuttal.

**On Nov 24 (Before the Information Leakage)**, Reviewer 2RNv accepted our clarifications and explicitly commented:

> *"Thanks for the detailed response, which clarifies many misunderstandings. **I have raised my score.** ... “*
>

**2) Conditional Acceptance from Reviewer s2Bn (Score 4)**

Reviewer s2Bn (Score 4) explicitly stated a **willingness to accept**:

> "Given these clarifications in an author response, **I would be willing to increase the score.**"
>

### 3. Summary of Rebuttals & Additional Evidence

During the rebuttal period, we provided **comprehensive additional analyses** to substantiate the robustness of our framework further. These experiments consistently confirmed the validity of our original claims:

**1) Validating Competitiveness against SOTA Baselines**

- **Borzoi Comparison:** Addressing R2RNv's request, we compared against Borzoi (a successor to Enformer). Results confirm CDBridge significantly outperforms it on isoform-level tasks. (Reviewer 2RNv Q9)
- **Linear Baselines:** We included Simple Linear and Evo2+Linear baselines, confirming that deep, context-aware modeling is essential ($R^2$  0.387 vs 0.021). (Reviewer 2RNv Q10)

**2) Confirming Robustness & Generalization**

- **Sequence Length:** We conducted a "Train Short (<20k), Test Long (>40k)" analysis. The model maintains high accuracy (AUC 0.990) on long sequences, proving it does not overfit to length. (Reviewer s2Bn Q1)
- **Ablation Studies:** We verified the robustness of the Tissue Encoder (scGPT vs. scFoundation) and the learning paradigm of Tissue Cluster(Learned vs. Fixed), showing consistent performance gains. (Reviewer DM1S)
- **Dynamic Merge Ratio:** We added an ablation comparing Dynamic vs. Fixed merge ratios, validating our sampling strategy prevents overfitting. (Reviewer DSq4 Q4 & Reviewer DM1S Q1)

**3) Clarifications**

- **GTEx-Benchmark Details:** We provided expanded details on splitting strategies and tissue balance to ensure reproducibility. (Reviewer DSq4 Q1 & Reviewer s2Bn Q1)
- **Clarified Metrics:** We corrected ambiguities in the ablation table regarding segmentation metrics when DNA input is disabled. (Reviewer 2RNv Q5 & Reviewer DM1S Q6)
- **Table 2 Reformatting:** We restructured Table 2 to strictly separate expression baselines from sequence encoder references, resolving the primary concern regarding baseline fairness. (Reviewer 2RNv Final Comment)

We believe CDBridge offers a meaningful step forward in context-aware biological modeling, particularly through its biologically grounded design and the gap-filling GTEx-Benchmark.

We respectfully submit that the current scores in the system (2 and 4) do not fully capture the constructive consensus reached during the discussion, where Reviewer 2RNv explicitly raised their score to 6 and Reviewer s2Bn (Score 4) promised a raise if question solved (before the information leakage). Given this positive trajectory and the strong endorsement from other reviewers (Scores 6 and 8), we sincerely hope you will find that our work merits a place at ICLR 2026.

**We remain fully available to address any further questions or details you may need to assist your decision.**

Best Regards,

The authors.

---

### Meta-Review · Area_Chair_Zk9s · 2026-01-07

**Summary:**

This paper proposes CDBridge, a two-stage post-training “bridge” that combines pretrained DNA and protein foundation models with a tissue-conditioned decoder to predict isoform-level expression under varying tissue contexts, and introduces GTEx-Benchmark with a leave-tissue-out setting for zero-shot tissue generalization.

Reviews were initially mixed: one reviewer rejected due to concerns about evaluation protocol clarity and baseline fairness/coverage, while others found the approach biologically motivated and potentially impactful but requested clearer benchmarking and ablations.

**Reviewer Concerns:**

### Addressed by rebuttal / revision

- Clarified benchmark construction and fairness (GTEx-Benchmark split/tissue balance; Table 2 separation of baselines), and added multiple robustness checks (e.g., dynamic vs fixed merge ratio; short-train/long-test length analysis).

- Baseline coverage: authors added comparisons/analyses requested by the reject reviewer (including Borzoi comparison in revision/appendix) and additional ablations.

- Information leakage concern (tissue context): the paper includes a control where using tissue embedding alone yields near-zero R2, supporting that tissue embeddings act as conditioning rather than directly encoding target-gene signal.


### Still outstanding / partially outstanding

- Strength of claims vs evidence: some positioning statements (e.g., “versatile biological foundation model”) may read stronger than what is fully supported without careful qualification and clearer scope.

- Presentation and reproducibility polish: reviewers asked for clearer wording/consistency and explicit clarifications on certain appendix/table details

**Reviewer Scores:**

- Reviewer 2RNv: 2 -> 6 (accept as poster), assuming the reviewer is engaged given the reviewer reportedly acknowledged clarifications and raised score during discussion, and key requested additions were addressed in the revision.

- Reviewer s2Bn: unchanged at 4 or raised up to 6, assuming the reviewer is engaged and the authors' clarified benchmark details and added robustness analyses are incorporated cleanly.

- Reviewer DSq4: 6–7, modest bump if final text tightens claims and clearly documents protocols/ablations.

- Reviewer DM1S: 8, supportive since beginning

---

### Decision · Program_Chairs · 2026-01-26

Accept (Poster)